# Spatio-Temporal Variational Gaussian Processes

**Oliver Hamelijnck**[*]
The Alan Turing Institute /
University of Warwick
ohamelijnck@turing.ac.uk

**William J. Wilkinson**[*]
Aalto University
william.wilkinson@aalto.fi

**Niki A. Loppi**
NVIDIA
nloppi@nvidia.com

**Arno Solin**
Aalto University
arno.solin@aalto.fi

**Theodoros Damoulas**
The Alan Turing Institute /
University of Warwick
tdamoulas@turing.ac.uk

## Abstract

We introduce a scalable approach to Gaussian process inference that combines spatio-temporal filtering with natural gradient variational inference, resulting in a non-conjugate GP method for multivariate data that scales linearly with respect to time. Our natural gradient approach enables application of parallel filtering and smoothing, further reducing the temporal span complexity to be logarithmic in the number of time steps. We derive a sparse approximation that constructs a state-space model over a reduced set of spatial inducing points, and show that for separable Markov kernels the full and sparse cases exactly recover the standard variational GP, whilst exhibiting favourable computational properties. To further improve the spatial scaling we propose a mean-field assumption of independence between spatial locations which, when coupled with sparsity and parallelisation, leads to an efficient and accurate method for large spatio-temporal problems.

## 1 Introduction

Most real-world processes occur across space and time, exhibit complex dependencies, and are observed through noisy irregular samples. Take, for example, the task of modelling air pollution across a city. Such a task involves large amounts of noisy, partially-observed data with strong seasonal effects governed by weather, traffic, human movement, *etc*. This setting motivates a probabilistic perspective, allowing for the incorporation of prior knowledge and the quantification of uncertainty.

Gaussian processes (GPs, [38]) provide such a probabilistic modelling paradigm, but their inherent cubic computational scaling in the number of data, $N$, limits their applicability to spatio-temporal tasks. Arguably the most successful methods to address this issue are sparse GPs [37], which summarise the true GP posterior through a reduced set of $M$ *inducing points* and have dominant computational scaling $\mathcal{O}(NM^2)$, and spatio-temporal GPs [43], which rewrite the GP prior as a state-space model and use filtering to perform inference in $\mathcal{O}(Nd^3)$, where $d$ is the dimensionality of the state-space. Sparse GPs and spatio-temporal GPs have been combined by constructing a Markovian system in which a set of *spatial* inducing points are tracked over time [24, 51].

However, existing methods for spatio-temporal GPs make approximations to the prior conditional model [24] or do not exploit natural gradients [45], meaning they do not provide the same inference and learning results as state-of-the-art variational GPs [26] in the presence of non-conjugate likelihoods or sparsity, which has hindered their widespread adoption. We introduce *spatio-temporal*

---

[*]equal contribution

35th Conference on Neural Information Processing Systems (NeurIPS 2021).

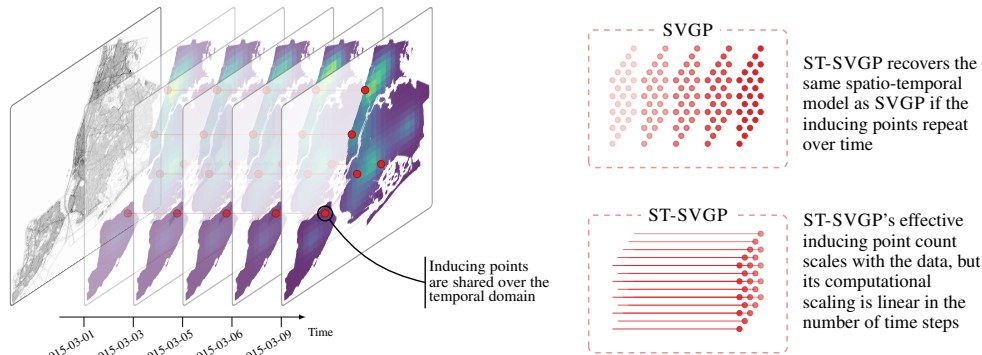

Figure 1: A demonstration of the spatio-temporal sparse variational GP (ST-SVGP) applied to crime count data in New York. ST-SVGP tracks spatial points over time via spatio-temporal filtering. The colourmap is the posterior mean, and the red dots are spatial inducing points. The diagram shows the difference between how inducing points are treated in ST-SVGP and SVGP.

*variational* GPs (ST-VGP), which provide the *exact* same results as standard variational GPs, whilst reducing the computational scaling in the temporal dimension from cubic to linear. ST-VGP is derived using a natural gradient variational inference approach based on filtering and smoothing. We also derive this method's sparse variant, and demonstrate how it enables the use of significantly more inducing points than the standard approach, leading to improved predictive performance.

We then show how the spatio-temporal structure can be exploited even further to improve both the temporal and spatial scaling. We demonstrate for the first time how to apply parallel filtering and smoothing [41] to non-conjugate GPs to reduce the temporal (span) complexity to be logarithmic. We then reformulate the model to enable an efficient mean-field approximation across space, improving the complexity with respect to the number of spatial points. We analyse the practical performance and scalability of our proposed methods, demonstrating how they make it possible to apply GPs to large-scale spatio-temporal scenarios without sacrificing inference quality.

## 1.1 Related Work

GPs are commonly used for spatio-temporal modelling in both machine learning and spatial statistics [38, 22, 14, 6]. Many approaches to overcome their computational burden have been proposed, from nearest neighbours [17] to parallel algorithms on GPUs [48]. Within machine learning, the sparse GP approach is perhaps the most popular [37, 46], and is typically combined with mini-batching to allow training on massive datasets [26]. However, it fails in practical cases where the number of inducing points must grow with the size of the data, such as for time series [49].

When the data lie on a grid, separable kernels exhibit Kronecker structure which can be exploited for efficient inference [39]. This approach has been generalised to the partial grid setting [53], and to structured kernel interpolation (SKI, [52]) which requires only that inducing points be on a grid. Generally, these approaches are limited to the conjugate case, although Laplace-based extensions exist [19]. Bruinsma et al. [10] present an approach to spatio-temporal modelling that performs an orthogonal projection of the data to enforce independence between the latent processes.

It has been shown that variational GPs can be computed in linear time either by exploiting sparse precision structure [18] or via filtering and smoothing [11]. Other inference schemes such as Laplace and expectation propagation have also been proposed [35, 51]. In the spatio-temporal case, sparsity has been used in the spatial dimension [24, 43]. These methods historically suffered from the fact that *i)* filtering was not amenable to fast automatic differentiation due to its recursive nature, and *ii)* state-of-the-art inference schemes had not been developed to make them directly comparable to other methods. The first is no longer an issue since many machine learning frameworks are now capable of efficiently differentiating recursive models [11]. We address the second point with this paper. A similar algorithm to ours that is also sparse in the temporal dimension has been developed [50, 2], and relevant properties of the spatio-temporal model presented here are also analysed in [45]. Fourier features [28] are an alternative approach to scalable GPs, but are not suited to very long time series with high variability due to the need for impractically many inducing features.

## 2 Background

We consider data lying on a spatio-temporal grid comprising input–output pairs, $\{\mathbf{X}^{(st)} \in \mathbb{R}^{N_t \times N_\mathbf{s} \times D}, \mathbf{Y}^{(st)} \in \mathbb{R}^{N_t \times N_\mathbf{s}}\}$, where $N_t$ is the number of temporal points, $N_\mathbf{s}$ the number of spatial points, and $D = 1 + D_\mathbf{s}$ the input dimensionality (with $D_\mathbf{s}$ being the number of spatial dimensions). We use $t$ and $\mathbf{s}$ to represent time and space respectively. The assumption of the grid structure is relaxed via the introduction of sparse methods in Sec. 3, and by the natural handling of missing data.

For consistency with the GP literature we let $\mathbf{X} = \text{vec}(\mathbf{X}^{(st)}) \in \mathbb{R}^{N \times D}$, $\mathbf{Y} = \text{vec}(\mathbf{Y}^{(st)}) \in \mathbb{R}^{N \times 1}$, where $N = N_t N_\mathbf{s}$ is the total number of data points. We use the operator $\text{vec}(\cdot)$ to simply convert data from a spatio-temporal grid into vector form, whilst keeping observations ordered by time and then space. For notational convenience we define $\mathbf{X}_{n,k} = \mathbf{X}_{n,k}^{(st)}$, $\mathbf{Y}_{n,k} = \mathbf{Y}_{n,k}^{(st)}$, which indexes data at time index $n$ and spatial index $k$. We use $t_n$ to denote the $n$'th time point, $\mathbf{S} \in \mathbb{R}^{N_\mathbf{s} \times D_\mathbf{s}}$ to denote all spatial grid points and $\mathbf{S}_k$ the $k$'th one. Let $f : \mathbb{R}^D \to \mathbb{R}$ to be a random function with a zero-mean GP prior, then for a given likelihood $p(\mathbf{Y} \mid f(\mathbf{X}))$ the generative model is,

$$f(x) \sim \mathcal{GP}(0, \kappa(x,x')), \quad \mathbf{Y} \mid \mathbf{f} \sim \prod_{n=1}^{N_t} \prod_{k=1}^{N_\mathbf{s}} p(\mathbf{Y}_{n,k} \mid \mathbf{f}_{n,k}), \tag{1}$$

where $\mathbf{f}_{n,k} = f(\mathbf{X}_{n,k})$, and we let $\mathbf{f}_n$ be the function values of all spatial points at time $t_n$. When the kernel $\kappa$ is evaluated at given inputs we write the corresponding gram matrix as $\mathbf{K}_{\mathbf{XX}'} = \kappa(\mathbf{X}, \mathbf{X}')$. To make it explicit that $f$ takes spatio-temporal inputs we also abuse the notation slightly to write $f(x) = f(t, \mathbf{s})$ and $\kappa(x,x') = \kappa(t, \mathbf{s}, t', \mathbf{s}')$. A summary of all notation used is provided in App. A. For Gaussian likelihoods the posterior, $p(\mathbf{f} \mid \mathbf{Y})$, is available in closed form, otherwise approximations must be used. In either case, inference typically comes at a cubic cost of $\mathcal{O}(N_t^3 N_\mathbf{s}^3)$.

### 2.1 State Space Spatio-Temporal Gaussian Processes

One method for handling the cubic scaling of GPs is to reformulate the prior in Eq. (1) as a state space model, reducing the computational scaling to linear in the number of time points [43]. The enabling assumption is that the kernel is both Markovian and separable between time and space: $\kappa(t, \mathbf{s}, t', \mathbf{s}') = \kappa_t(t, t') \kappa_\mathbf{s}(\mathbf{s}, \mathbf{s}')$. We use the term *Markovian kernel* to refer to a kernel which can be re-written in state-space form (see [44] for an overview). First, we write down the GP prior as a stochastic partial differential equation (SPDE, see [15]) $\partial_t \bar{\mathbf{f}}(t, \mathbf{s}) = \mathcal{A}_\mathbf{s} \bar{\mathbf{f}}(t, \mathbf{s}) + \mathcal{L}_\mathbf{s} \mathbf{w}(t, \mathbf{s})$, where $\mathbf{w}(t, \mathbf{s})$ is a (spatio-temporal) white noise process and $\mathcal{A}_\mathbf{s}$ a suitable (pseudo-)differential operator [see 42]. By appropriately defining the model matrices and the white noise spectral density function, SPDEs of this form can represent a large class of separable and non-separable GP models.

When the kernel is separable, this SPDE can be simplified to a finite-dimensional SDE [24] by marginalising to a finite set of spatial locations, $\mathbf{S} \in \mathbb{R}^{N_\mathbf{s} \times D_\mathbf{s}}$, giving, $d\bar{\mathbf{f}}(t) = \mathbf{F} \bar{\mathbf{f}}(t)\,dt + \mathbf{L}\,d\boldsymbol{\beta}(t)$, where $\bar{\mathbf{f}}(t)$ is the Gaussian distributed state at the spatial points $\mathbf{S}$ at time $t$, with dimensionality $d = N_\mathbf{s} d_t$, where $d_t$ is the dimensionality of the state-space model induced by $\kappa_t(\cdot, \cdot)$. $d\boldsymbol{\beta}(t)$ has spectral density $\mathbf{Q}_c$, and the matrix $\mathbf{H}$ extracts the function value from the state: $\mathbf{f}_n = \mathbf{H}\bar{\mathbf{f}}(t_n)$. $\mathbf{F}$ and $\mathbf{L}$ are the feedback and noise effect matrices [42]. This simplification to an SDE is possible due to the independence between spatial points at time $t$ and all other time steps, given the current state [45]. This follows from the fact that for *any* separable kernel, $f(t, \mathbf{s})$ and $f(t', \mathbf{s}')$ are independent given $f(t', \mathbf{s})$ [36]. For a step size $\Delta_n = t_{n+1} - t_n$, the discrete-time model matrices are,

$$\mathbf{A}_n = \mathbf{\Phi}(\mathbf{F}\Delta_n), \qquad \mathbf{Q}_n = \int_0^{\Delta_n} \mathbf{\Phi}(\Delta_n - \tau)\, \mathbf{L}\, \mathbf{Q}_c\, \mathbf{L}^\top\, \mathbf{\Phi}(\Delta_n - \tau)^\top\, d\tau, \tag{2}$$

where $\mathbf{\Phi}(\cdot)$ is the matrix exponential. The resulting discrete model is,

$$\bar{\mathbf{f}}(t_{n+1}) = \mathbf{A}_n \bar{\mathbf{f}}(t_n) + \mathbf{q}_n, \qquad \mathbf{Y}_n \mid \bar{\mathbf{f}}(t_n) \sim p(\mathbf{Y}_n \mid \mathbf{H}\bar{\mathbf{f}}(t_n)), \tag{3}$$

where $\mathbf{q}_n \sim \text{N}(\mathbf{0}, \mathbf{Q}_n)$. If $p(\mathbf{Y}_n \mid \mathbf{H}\bar{\mathbf{f}}(t_n))$ is Gaussian then Kalman smoothing algorithms can be employed to perform inference in Eq. (3) in $\mathcal{O}(N_t d^3) = \mathcal{O}(N_t N_\mathbf{s}^3 d_t^3)$.

**Markovian GPs with Spatial Sparsity** Sparse GPs re-define the GP prior over a smaller set of $M$ *inducing points*: let $\mathbf{u} = f(\mathbf{Z}) \in \mathbb{R}^{M \times 1}$ be the inducing variables at inducing locations $\mathbf{Z} \in \mathbb{R}^{M \times D}$, then the augmented prior is $p(\mathbf{f}, \mathbf{u}) = p(\mathbf{f} \mid \mathbf{u})p(\mathbf{u})$, where $p(\mathbf{u}) = \text{N}(\mathbf{u} \mid \mathbf{0}, \mathbf{K}_{\mathbf{ZZ}})$, and with Gaussian conditional $p(\mathbf{f} \mid \mathbf{u})$. If the inducing points are placed on a spatio-temporal grid, with $\mathbf{Z}_\mathbf{s} \in \mathbb{R}^{M_\mathbf{s} \times D_\mathbf{s}}$ being the spatial inducing locations, the conditional $p(\mathbf{f} \mid \mathbf{u})$ can be simplified to (see App. D):

$$p(\mathbf{f} \mid \mathbf{u}) = \text{N}\big(\mathbf{f} \mid \big[\mathbf{I} \otimes (\mathbf{K}_{\mathbf{SS}}^{(\mathbf{s})} \otimes \mathbf{K}_{\mathbf{Z}_\mathbf{s}\mathbf{Z}_\mathbf{s}}^{-(\mathbf{s})})\big]\mathbf{u}, \mathbf{K}_{tt}^{(t)} \otimes \widetilde{\mathbf{Q}}_\mathbf{s}\big), \tag{4}$$

where $\widetilde{\mathbf{Q}}_\mathbf{s} = \mathbf{K}_{\mathbf{SS}}^{(\mathbf{s})} - \mathbf{K}_{\mathbf{SZ_s}}^{(\mathbf{s})} \mathbf{K}_{\mathbf{Z_sZ_s}}^{-(\mathbf{s})} \mathbf{K}_{\mathbf{Z_sS}}^{(\mathbf{s})}$ (see App. A for notational details). The *fully independent training conditional* (FITC) method [37] approximates the full conditional covariance with its diagonal, leading to the following convenient property: $q_{\text{FITC}}(\mathbf{f} \,|\, \mathbf{u}) = \prod_{n=1}^{N_t} q_{\text{FITC}}(\mathbf{f}_n \,|\, \mathbf{u}) = \prod_{n=1}^{N_t} q_{\text{FITC}}(\mathbf{f}_n \,|\, \mathbf{u}_n)$, where the last equality holds because $\mathbf{I} \otimes (\mathbf{K}_{\mathbf{SS}}^{(\mathbf{s})} \otimes \mathbf{K}_{\mathbf{Z_sZ_s}}^{-(\mathbf{s})})$ is block diagonal. This factorisation across time allows the model to be cast into the state-space form of Eq. (3), but where the state $\bar{\mathbf{f}}(t)$ is defined over the reduced set of spatial inducing points [24]. Inference can be performed in $\mathcal{O}(N_t M_\mathbf{s}^3 d_t^3)$.

## 2.2 Sparse Variational GPs

To perform approximate inference in the presence of sparsity or non-Gaussian likelihoods, variational methods cast inference as optimisation through minimisation of the Kullback–Leibler divergence (KLD) from the true posterior to the approximate posterior [8]. Although direct computation of the KLD is intractable, it can be rewritten as the maximisation of the evidence lower bound (ELBO).

Unlike FITC, the sparse variational GP (SVGP, [46]) does not approximate the conditional $p(\mathbf{f} \,|\, \mathbf{u})$, but instead approximates the posterior as $q(\mathbf{f}, \mathbf{u}) = p(\mathbf{f} \,|\, \mathbf{u}) \, q(\mathbf{u})$, where $q(\mathbf{u}) = \mathrm{N}(\mathbf{u} \,|\, \mathbf{m}, \mathbf{P})$ is a Gaussian whose parameters are to be optimised. The SVGP ELBO is:

$$\mathcal{L}_{\text{SVGP}} = \mathbb{E}_{q(\mathbf{u})} \left[ \mathbb{E}_{q(\mathbf{f} \,|\, \mathbf{u})} \left[ \log p(\mathbf{Y} \,|\, \mathbf{f}) \right] \right] - \mathrm{KL} \left[ q(\mathbf{u}) \,\|\, p(\mathbf{u}) \right], \tag{5}$$

which can be computed in $\mathcal{O}(NM^2 + M^3)$. SVGP has many benefits over methods such as FITC, including: non-Gaussian likelihoods can be handled through quadrature or Monte-Carlo approximations [27, 33], it is applicable to big data through stochastic VI and mini-batching [26], and the inducing locations are 'variationally protected' and hence prevent overfitting [7].

**Natural Gradients** Natural gradient descent calculates gradients in *distribution* space rather than *parameter* space, and has been shown to improve inference time and quality for variational GPs [26, 40]. A natural descent direction is obtained by scaling the standard gradient by the inverse of the Fisher information matrix, $\mathbb{E}_{q(\cdot)} \left[ \nabla^2 \log q(\cdot) \right]$ [5]. For a Gaussian approximate posterior, the natural gradient of target $\mathcal{L}$ with respect to the natural parameters $\boldsymbol{\lambda}$ can be calculated without directly forming the Hessian, since it can be shown to be equivalent to the gradient with respect to the mean parameters $\boldsymbol{\mu} = [\mathbf{m}, \mathbf{m}\mathbf{m}^\top + \mathbf{P}]$ [25]. The natural parameter update, with learning rate $\beta$, becomes,

$$\boldsymbol{\lambda} \leftarrow \boldsymbol{\lambda} + \beta \, \frac{\partial \mathcal{L}}{\partial \boldsymbol{\mu}}. \tag{6}$$

To update the approximation posterior, $\boldsymbol{\lambda}$ can be simply transformed to the moment parameterisation $[\mathbf{m}, \mathbf{P}]$. A table of mappings between the various parametrisations is given in App. G.

**CVI and the Approximate Likelihood** Khan and Lin [31] show that when the prior and approximate posterior are conjugate (as in the GP case), further elegant properties of exponential family distributions mean that Eq. (6) is equivalent to a two step Bayesian update:

$$\widetilde{\boldsymbol{\lambda}} \leftarrow (1 - \beta) \, \widetilde{\boldsymbol{\lambda}} + \beta \, \frac{\partial \, \mathbb{E}_{q(\mathbf{f})} [\log p(\mathbf{Y} \,|\, \mathbf{f})]}{\partial \boldsymbol{\mu}}, \qquad \boldsymbol{\lambda} \leftarrow \boldsymbol{\eta} + \widetilde{\boldsymbol{\lambda}}, \tag{7}$$

where $\boldsymbol{\eta}$ are the natural parameters of the prior and $\widetilde{\boldsymbol{\lambda}}$ are the natural parameters of the likelihood contribution. Letting $\mathbf{g}(\cdot) = \frac{\partial \mathbb{E}_q [\log p(\mathbf{Y} \,|\, \mathbf{f})]}{\partial \cdot}$, the gradients can be computed in terms of the mean and covariance via the chain rule: $\mathbf{g}(\boldsymbol{\mu}) = \left[ \mathbf{g}(\mathbf{m}) - 2 \, \mathbf{g}(\mathbf{P}) \, \mathbf{m}, \; \mathbf{g}(\mathbf{P}) \right]$. Eq. (7) shows that, since the prior parameters $\boldsymbol{\eta}$ are known, natural gradient variational inference is completely characterised by updates to an *approximate likelihood*, which we denote $\mathrm{N}(\widetilde{\boldsymbol{Y}} \,|\, \mathbf{f}, \widetilde{\mathbf{V}})$, parameterised by covariance $\widetilde{\mathbf{V}} = (-2\widetilde{\boldsymbol{\lambda}}^{(2)})^{-1}$ and mean $\widetilde{\boldsymbol{Y}} = \widetilde{\mathbf{V}} \widetilde{\boldsymbol{\lambda}}^{(1)}$ (see App. A). The approximate posterior then has the form,

$$q(\mathbf{f}) = \frac{\mathrm{N}(\widetilde{\boldsymbol{Y}} \,|\, \mathbf{f}, \widetilde{\mathbf{V}}) \, p(\mathbf{f})}{\int \mathrm{N}(\widetilde{\boldsymbol{Y}} \,|\, \mathbf{f}, \widetilde{\mathbf{V}}) \, p(\mathbf{f}) \, \mathrm{d}\mathbf{f}}. \tag{8}$$

Computing $q(\mathbf{f})$ amounts to performing conjugate GP regression with the model prior and the approximation likelihood. This approach is called conjugate-computation variational inference (CVI, [31]). To re-emphasise that the CVI updates compute the exact same quantity as Eq. (6), we provide an alternative derivation in App. H by directly applying the chain rule to Eq. (6).

# 3 Spatio-Temporal Variational Gaussian Processes

In this section we introduce a spatio-temporal VGP that has linear complexity with respect to time whilst obtaining the identical variational posterior as the standard VGP. We will then go on to derive this method's sparse variant, which gives the same posterior as SVGP when the inducing points are set similarly (*i.e.*, on a spatio-temporal grid), but is capable of scaling to much larger values of $M$.

## 3.1 The Spatio-Temporal VGP ELBO

We first derive our proposed spatio-temporal VGP ELBO. We do this by exploiting the form of the approximate posterior after a natural gradient step in order to write the ELBO as a sum of three terms, each of which can be efficiently computed through filtering and smoothing. As shown in Sec. 2.2, after a natural gradient step, the approximate posterior $q(\mathbf{f}) \propto \mathrm{N}(\widetilde{\mathbf{Y}} \,|\, \mathbf{f}, \widetilde{\mathbf{V}})\, p(\mathbf{f})$ decomposes as a Bayesian update applied to the model prior with an approximate likelihood. Following Chang et al. [11] we substitute Eq. (8) into the VGP ELBO:

$$\mathcal{L}_{\text{VGP}} = \mathbb{E}_{q(\mathbf{f})} \left[ \log \frac{p(\mathbf{Y} \,|\, \mathbf{f})\, p(\mathbf{f})}{q(\mathbf{f})} \right] = \mathbb{E}_{q(\mathbf{f})} \left[ \log \frac{p(\mathbf{Y} \,|\, \mathbf{f})\, \cancel{p(\mathbf{f})} \int \mathrm{N}(\widetilde{\mathbf{Y}} \,|\, \mathbf{f}, \widetilde{\mathbf{V}})\, p(\mathbf{f})\, \mathrm{d}\mathbf{f}}{\mathrm{N}(\widetilde{\mathbf{Y}} \,|\, \mathbf{f}, \widetilde{\mathbf{V}})\, \cancel{p(\mathbf{f})}} \right] \tag{9}$$

$$= \sum_{n=1}^{N_t} \sum_{k=1}^{N_s} \mathbb{E}_{q(\mathbf{f}_{n,k})} \big[ \log p(\mathbf{Y}_{n,k} \,|\, \mathbf{f}_{n,k}) \big] - \mathbb{E}_{q(\mathbf{f})} \big[ \log \mathrm{N}(\widetilde{\mathbf{Y}} \,|\, \mathbf{f}, \widetilde{\mathbf{V}}) \big] + \log \mathbb{E}_{p(\mathbf{f})} \big[ \mathrm{N}(\widetilde{\mathbf{Y}} \,|\, \mathbf{f}, \widetilde{\mathbf{V}}) \big].$$

The first term is the expected log likelihood, the second is the expected log *approximate likelihood*, and the final term is the log marginal likelihood of the approximation posterior, $\log p(\widetilde{\mathbf{Y}}) = \log \mathbb{E}_{p(\mathbf{f})} \big[ \mathrm{N}(\widehat{\mathbf{Y}} \,|\, \mathbf{f}, \widetilde{\mathbf{V}}) \big]$. Naïvely evaluating $\mathcal{L}_{\text{VGP}}$ requires $\mathcal{O}(N^3)$ computation for both the update to $q(\mathbf{f})$ and the marginal likelihood. We now show how to compute this with linear scaling in $N_t$.

We observe that after a natural gradient update, $\widetilde{\mathbf{V}}$, the approximate likelihood covariance, has the same form as the gradient $\mathbf{g}(\mathbf{P})$ because, as seen in Eq. (7), $\widetilde{\boldsymbol{\lambda}}$ is only additively updated by $\mathbf{g}(\boldsymbol{\mu})$. Since the expected likelihood, $\mathbb{E}_{q(\mathbf{f})}[\log p(\mathbf{Y} \,|\, \mathbf{f})]$, factorises across observations, $\mathbf{g}(\mathbf{P})$ is diagonal and hence so is $\widetilde{\mathbf{V}}$. The approximate likelihood therefore factorises in the same way as the true one:

$$\log \mathrm{N}(\widetilde{\mathbf{Y}} \,|\, \mathbf{f}, \widetilde{\mathbf{V}}) = \sum_{n=1}^{N_t} \sum_{k=1}^{N_s} \log \mathrm{N}(\widetilde{\mathbf{Y}}_{n,k} \,|\, \mathbf{f}_{n,k}, \widetilde{\mathbf{V}}_{n,k}). \tag{10}$$

We now turn our attention to computing the posterior and the marginal likelihood. Recall that if the kernel is separable between time and space, $\kappa(t, \mathbf{s}, t', \mathbf{s}') = \kappa_t(t, t')\, \kappa_{\mathbf{s}}(\mathbf{s}, \mathbf{s}')$, then the GP prior can be re-written as Eq. (3). This separability property further results in the state-space model matrices having a convenient Kronecker structure,

$$\bar{\mathbf{f}}(t_{n+1}) = \left[ \mathbf{I}_{N_s} \otimes \mathbf{A}_n^{(t)} \right] \bar{\mathbf{f}}(t_n) + \mathbf{q}_n, \qquad \widetilde{\mathbf{Y}}_n \,|\, \bar{\mathbf{f}}(t_n) \sim p(\widetilde{\mathbf{Y}}_n \,|\, \mathbf{H}\, \bar{\mathbf{f}}(t_n)), \tag{11}$$

where $\mathbf{q}_n \sim \mathrm{N}(\mathbf{0}, \mathbf{K}_{\mathbf{SS}}^{(\mathbf{s})} \otimes \mathbf{Q}_n^{(t)})$ and $\mathbf{H} = \mathbf{I}_{N_s} \otimes \mathbf{H}^{(t)}$. Here $\mathbf{A}_n^{(t)} \in \mathbb{R}^{d_t \times d_t}$, $\mathbf{Q}_n^{(t)} \in \mathbb{R}^{d_t \times d_t}$, and $\mathbf{H}^{(t)} \in \mathbb{R}^{1 \times d_t}$ are the transition matrix, process noise covariance, and measurement model of the SDE (see Sec. 2.1) induced by the kernel $\kappa_t(\cdot, \cdot)$, respectively.

Because the GP prior is Markov and the approximate likelihood factorises across time, the approximate GP posterior is also Markov [45]. Hence marginals $q(\mathbf{f}_n)$ can be computed through linear filtering and smoothing applied to Eq. (11). Furthermore, the marginal likelihood of a linear Gaussian state-space model, $p(\widetilde{\mathbf{Y}}) = p(\widetilde{\mathbf{Y}}_1) \prod_{n=2}^{N_t} p(\widetilde{\mathbf{Y}}_n \,|\, \widetilde{\mathbf{Y}}_{1:n-1})$, can be computed sequentially by running the forward filter, since $p(\widetilde{\mathbf{Y}}_n \,|\, \widetilde{\mathbf{Y}}_{1:n-1}) = \int p(\widetilde{\mathbf{Y}}_n \,|\, \mathbf{H}\bar{\mathbf{f}}(t_n))\, p(\bar{\mathbf{f}}(t_n) \,|\, \widetilde{\mathbf{Y}}_{1:n-1})\, \mathrm{d}\bar{\mathbf{f}}(t_n)$, where $p(\bar{\mathbf{f}}(t_n) \,|\, \widetilde{\mathbf{Y}}_{1:n-1})$ is the predictive filtering distribution. By combining all of the above properties we can now write the ELBO as,

$$\mathcal{L}_{\text{ST-VGP}} = \sum_{n=1}^{N_t} \sum_{k=1}^{N_s} \mathbb{E}_{q(\mathbf{f}_{n,k})} \big[ \log p(\mathbf{Y}_{n,k} \,|\, \mathbf{f}_{n,k}) \big] - \sum_{n=1}^{N_t} \sum_{k=1}^{N_s} \mathbb{E}_{q(\mathbf{f}_{n,k})} \big[ \log \mathrm{N}(\widetilde{\mathbf{Y}}_{n,k} \,|\, \mathbf{f}_{n,k}, \widetilde{\mathbf{V}}_{n,k}) \big]$$

$$+ \sum_{n=1}^{N_t} \log \mathbb{E}_{p(\bar{\mathbf{f}}(t_n) \,|\, \widetilde{\mathbf{Y}}_{1:n-1})} \big[ \mathrm{N}(\widetilde{\mathbf{Y}}_n \,|\, \mathbf{H}\bar{\mathbf{f}}(t_n), \widetilde{\mathbf{V}}_n) \big]. \tag{12}$$

| **Algorithm 1** Spatio-temporal sparse VGP | **Algorithm 2** Sparse spatio-temporal smoothing |
|---|---|
| **Input:** Data:$\{\mathbf{X}, \mathbf{Y}\}$, Initial params.:$\{\widetilde{\mathbf{Y}}, \widetilde{\mathbf{V}}\}$, Learning rates:$\{\beta, \rho\}$ 
 **while** ELBO ($\mathcal{L}$) not converged **do** 
 $\quad$ ▷ *CVI natural gradient step:* 
 $\quad q(\mathbf{u}), \ell = $ Alg. 2$(\widetilde{\mathbf{Y}}, \widetilde{\mathbf{V}})$ 
 $\quad \mathcal{E} = \mathbb{E}_{q(\mathbf{u})}[\mathbb{E}_{p(\mathbf{f}\,|\,\mathbf{u})}[\log[p(\mathbf{Y}\,|\,\mathbf{f})]]$ 
 $\quad \widetilde{\boldsymbol{\lambda}} = (1 - \beta)\widetilde{\boldsymbol{\lambda}} + \beta \frac{\partial \mathcal{E}}{\partial \boldsymbol{\mu}}$ 
 $\quad \widetilde{\mathbf{V}} = (-2\widetilde{\boldsymbol{\lambda}}^{(2)})^{-1}, \quad \widetilde{\mathbf{Y}} = \widetilde{\mathbf{V}}\widetilde{\boldsymbol{\lambda}}^{(1)}$ 
 $\quad$ ▷ *Hyperparameter gradient step:* 
 $\quad q(\mathbf{u}), \ell = $ Alg. 2$(\widetilde{\mathbf{Y}}, \widetilde{\mathbf{V}})$ 
 $\quad \mathcal{E} = \mathbb{E}_{q(\mathbf{u})}[\mathbb{E}_{p(\mathbf{f}\,|\,\mathbf{u})}[\log[p(\mathbf{Y}\,|\,\mathbf{f})]]$ 
 $\quad \mathcal{L} = \mathcal{E} - \mathbb{E}_{q(\mathbf{u})}[\log \mathrm{N}(\widetilde{\mathbf{Y}}\,|\,\mathbf{u}, \widetilde{\mathbf{V}})] + \ell \quad$ ▷ *ELBO* 
 $\quad \boldsymbol{\theta} = \boldsymbol{\theta} + \rho \frac{\partial \mathcal{L}}{\partial \boldsymbol{\theta}}$ 
 **end while** | **Input:** Likelihood:$\{\widetilde{\mathbf{Y}}, \widetilde{\mathbf{V}}\}$, Space prior:$\{\mathbf{K}_{\mathbf{Z_s Z_s}}^{(\mathbf{s})}\}$, Time prior:$\{\mathbf{A}^{(t)}, \mathbf{Q}^{(t)}, \mathbf{H}^{(t)}\}$ 
 ▷ *Construct model matrices:* 
 $\mathbf{A}_n = \mathbf{I}_{M_s} \otimes \mathbf{A}_n^{(t)}$, 
 $\mathbf{Q}_n = \mathbf{K}_{\mathbf{Z_s Z_s}}^{(\mathbf{s})} \otimes \mathbf{Q}_n^{(t)}$ , 
 $\mathbf{H} = \mathbf{I}_{M_s} \otimes \mathbf{H}^{(t)}$ 
 ▷ *Filtering and smoothing:* 
 **if** using parallel filter / smoother **then** 
 $\quad q(\mathbf{u}), \ell = $ Alg. 4$(\widetilde{\mathbf{Y}}, \widetilde{\mathbf{V}}, \mathbf{A}, \mathbf{Q}, \mathbf{H})$ 
 **else** 
 $\quad q(\mathbf{u}), \ell = $ Alg. 3$(\widetilde{\mathbf{Y}}, \widetilde{\mathbf{V}}, \mathbf{A}, \mathbf{Q}, \mathbf{H})$ 
 **end if** 
 ▷ *Return posterior marginals and log likelihood:* 
 **return** $q(\mathbf{u}), \ell$ |

This ELBO can be computed with linear scaling in $N_t$: $\mathcal{O}(N_t N_\mathbf{s}^3 d_t^3)$. We now show that the natural gradient step for updating the parameters of $\mathrm{N}(\widetilde{\mathbf{Y}}\,|\,\mathbf{f}, \widetilde{\mathbf{V}})$ can be computed with the same complexity.

### 3.2 Efficient Natural Gradient Updates

As discussed in Sec. 2.2, a *natural gradient* update to the posterior, $q(\mathbf{f}) \propto p(\mathbf{f}) \mathrm{N}(\widetilde{\mathbf{Y}}\,|\,\mathbf{f}, \widetilde{\mathbf{V}})$, has superior convergence properties to gradient descent, and is completely characterised by an update to the approximate likelihood, $\mathrm{N}(\widetilde{\mathbf{Y}}\,|\,\mathbf{f}, \widetilde{\mathbf{V}})$, whose mean and covariance are the free parameters of the model, and implicitly define the same *variational* parameters as VGP. Since the likelihood factorises across the data points, these updates only require computation of the marginal distribution $q(\mathbf{f}_{n,k})$ to obtain $\mathbb{E}_{q(\mathbf{f}_{n,k})}[\log p(\mathbf{Y}_{n,k}\,|\,\mathbf{f}_{n,k})]$ and its gradients.

As we have shown, computation of the marginal posterior amounts to smoothing over the state, $\bar{\mathbf{f}} \sim \mathrm{N}(\bar{\mathbf{f}}\,|\,\bar{\boldsymbol{m}}, \bar{\mathbf{P}})$, with the model in Eq. (11). The time marginals are given by applying the measurement model to the state: $q(\mathbf{f}_n) = \mathrm{N}(\mathbf{f}_n\,|\,\mathbf{m}_n = \mathbf{H}\bar{\boldsymbol{m}}_n, \mathbf{P}_n = \mathbf{H}\bar{\mathbf{P}}_n\mathbf{H}^\top)$ after which $q(\mathbf{f}_{n,k}) = \int q(\mathbf{f}_n)\, \mathrm{d}\mathbf{f}_{n,\neq k}$ can then be obtained by integrating out the other spatial points. Given the marginal, Eq. (7) can be used to give the new likelihood parameters $\widetilde{\mathbf{Y}}$ and $\widetilde{\mathbf{V}}$. The full learning algorithm iterates this process alternately with a hyperparameter update via gradient descent applied to the ELBO, Eq. (12), and has computational complexity $\mathcal{O}(N_t N_\mathbf{s}^3 d_t^3)$. We call this method the *spatio-temporal variational GP* (ST-VGP).

### 3.3 Spatial Sparsity: from $\mathcal{O}(N_t N_\mathbf{s}^3 d_t^3)$ to $\mathcal{O}(N_t M_\mathbf{s}^3 d_t^3)$

We now introduce spatial inducing points, $\mathbf{Z_s}$, in order to reduce the effective dimensionality of the state-space model. Whilst we maintain the same notation for consistency, it should be noted that the sparse model no longer requires the data to be on a spatio-temporal grid, only that the inducing points are. In this case, letting $q(\mathbf{u}) = \mathrm{N}(\mathbf{u}\,|\,\mathbf{m}^{(\mathbf{u})}, \mathbf{P}^{(\mathbf{u})})$ be the sparse variational posterior, the marginal $q(\mathbf{f}_n) = \mathrm{N}(\mathbf{f}\,|\,\mathbf{m}_n, \mathbf{P}_n)$ only depends on $\mathbf{m}_n^{(\mathbf{u})}, \mathbf{P}_n^{(\mathbf{u})}$ due to the conditional independence property for separable kernels discussed in Sec. 2.1. We compute the posterior $q(\mathbf{u})$ via filtering and smoothing over the state $\bar{\mathbf{f}}(t)$ in a similar way to ST-VGP by setting,

$$\mathbf{A}_n = \mathbf{I}_{M_s} \otimes \mathbf{A}_n^{(t)}, \qquad \mathbf{Q}_n = \mathbf{K}_{\mathbf{Z_s Z_s}}^{(\mathbf{s})} \otimes \mathbf{Q}_n^{(t)}, \qquad \mathbf{H} = \mathbf{I}_{M_s} \otimes \mathbf{H}^{(t)}. \tag{13}$$

Alg. 2 gives the smoothing algorithm. However, the natural gradient update, Eq. (7), now becomes,

$$\widetilde{\boldsymbol{\lambda}} \leftarrow (1 - \beta)\,\widetilde{\boldsymbol{\lambda}} + \beta\, \frac{\partial\, \mathbb{E}_{q(\mathbf{u})}\left[\mathbb{E}_{p(\mathbf{f}\,|\,\mathbf{u})}\left[\log p(\mathbf{Y}\,|\,\mathbf{f})\right]\right]}{\partial \boldsymbol{\mu}^{(\mathbf{u})}}, \tag{14}$$

which results in $\widetilde{\boldsymbol{\lambda}}_n^{(2)}$, and hence also $\widetilde{\mathbf{V}}_n$, being a dense matrix (*i.e.*, $\widetilde{\mathbf{V}}$ is block-diagonal) due to the conditional mapping, $p(\mathbf{f}_n\,|\,\mathbf{u}_n)$. Therefore the approximate likelihood for the sparse model factorises across time, but not space (see App. J for details): $\log \mathrm{N}(\widetilde{\mathbf{Y}}\,|\,\mathbf{u}, \widetilde{\mathbf{V}}) = \sum_{n=1}^{N_t} \log \mathrm{N}(\widetilde{\mathbf{Y}}_n\,|\,\mathbf{u}_n, \widetilde{\mathbf{V}}_n)$.

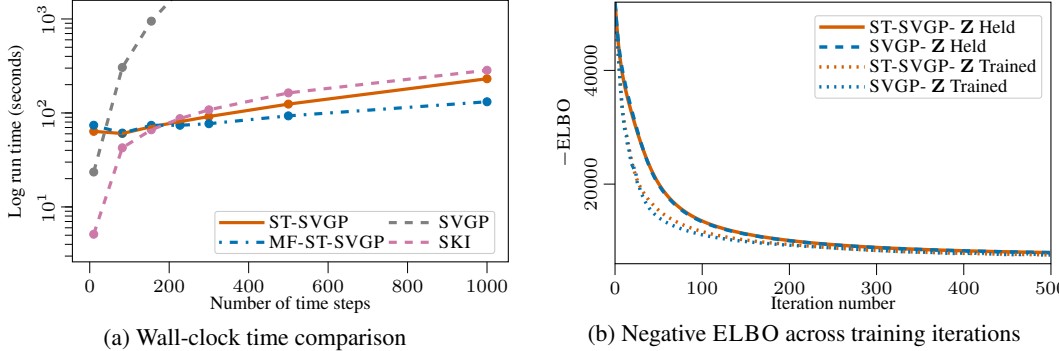

(a) Wall-clock time comparison          (b) Negative ELBO across training iterations

Figure 2: (a) Log wall-clock time, including any startup costs, across 7 synthetic spatio-temporal datasets with an increasing number of time steps (average across 5 runs). (b) Negative ELBO during training for the small-scale NYC-CRIME dataset.

**The Spatio-Temporal Sparse VGP ELBO**   Adding inducing points in space is equivalent to placing the inducing points on a spatio-temporal grid (*i.e.*, inducing points exist at all time steps), and hence the variational objective directly follows from $\mathcal{L}_{\text{SVGP}}$ using a similar argument to Sec. 3.1:

$$
\mathcal{L}_{\text{ST-SVGP}} = \mathbb{E}_{q(\mathbf{f},\mathbf{u})}\left[\log \frac{p(\mathbf{Y}\mid\mathbf{f})\,\cancel{p(\mathbf{f}\mid\mathbf{u})}\,\cancel{p(\mathbf{u})}\int \text{N}(\widetilde{\mathbf{Y}}\mid\mathbf{u},\widetilde{\mathbf{V}})\,p(\mathbf{u})\,\mathrm{d}\mathbf{u}}{\text{N}(\widetilde{\mathbf{Y}}\mid\mathbf{u},\widetilde{\mathbf{V}})\,\cancel{p(\mathbf{f}\mid\mathbf{u})}\,\cancel{p(\mathbf{u})}}\right]
$$

$$
= \sum_{n=1}^{N_t}\sum_{k=1}^{N_s}\mathbb{E}_{q(\mathbf{u}_n)}\big[\mathbb{E}_{p(\mathbf{f}_{n,k}\mid\mathbf{u}_n)}\big[\log p(\mathbf{Y}_{n,k}\mid\mathbf{f}_{n,k})\big]\big] - \sum_{n=1}^{N_t}\mathbb{E}_{q(\mathbf{u}_n)}\big[\log \text{N}(\widetilde{\mathbf{Y}}_n\mid\mathbf{u}_n,\widetilde{\mathbf{V}}_n)\big]
$$

$$
+ \sum_{n=1}^{N_t}\log \mathbb{E}_{p(\bar{\mathbf{f}}(t_n)\mid\widetilde{\mathbf{Y}}_{1:n-1})}\big[\text{N}(\widetilde{\mathbf{Y}}_n\mid\mathbf{H}\bar{\mathbf{f}}(t_n),\widetilde{\mathbf{V}}_n)\big], \tag{15}
$$

where the final term is given by the forward filter.

**Efficient Natural Gradient Updates**   The marginal required to compute the ELBO and natural gradient, $q(\mathbf{f}_{n,k}) = \iint p(\mathbf{f}\mid\mathbf{u})\,q(\mathbf{u})\,\mathrm{d}\mathbf{u}\,\mathrm{d}\mathbf{f}_{\neq n,k} = \iint p(\mathbf{f}\mid\mathbf{u}_n)\,q(\mathbf{u}_n)\,\mathrm{d}\mathbf{u}_n\,\mathrm{d}\mathbf{f}_{\neq n,k}$, is the predictive distribution at input $\mathbf{X}_{n,k}$ from the posterior $q(\mathbf{u})$. Because the inducing points have only been placed in space, this can be simplified through the Kronecker structure given by the state-space model. As shown in App. I, the marginal mean and covariance are,

$$
\mathbf{m}_{n,k} = \mathbf{K}^{(\mathbf{s})}_{\mathbf{S}_k\mathbf{Z}_\mathbf{s}}\mathbf{K}^{-(\mathbf{s})}_{\mathbf{Z}_\mathbf{s}\mathbf{Z}_\mathbf{s}}\mathbf{m}^{(\mathbf{u})}_n,
$$

$$
\mathbf{P}_{n,k} = \mathbf{K}^{(\mathbf{s})}_{\mathbf{S}_k\mathbf{Z}_\mathbf{s}}\mathbf{K}^{-(\mathbf{s})}_{\mathbf{Z}_\mathbf{s}\mathbf{Z}_\mathbf{s}}\mathbf{P}^{(\mathbf{u})}_n\mathbf{K}^{-(\mathbf{s})}_{\mathbf{Z}_\mathbf{s}\mathbf{Z}_\mathbf{s}}\mathbf{K}^{(\mathbf{s})}_{\mathbf{Z}_\mathbf{s}\mathbf{S}_k} + \mathbf{K}^{(t)}_{\mathbf{X}_{n,k}\mathbf{X}_{n,k}}\big(\mathbf{K}^{(\mathbf{s})}_{\mathbf{S}_k\mathbf{S}_k} - \mathbf{K}^{(\mathbf{s})}_{\mathbf{S}_k\mathbf{Z}_\mathbf{s}}\mathbf{K}^{-(\mathbf{s})}_{\mathbf{Z}_\mathbf{s}\mathbf{Z}_\mathbf{s}}\mathbf{K}^{(\mathbf{s})}_{\mathbf{Z}_\mathbf{s}\mathbf{S}_k}\big), \tag{16}
$$

where $\mathbf{m}^{(\mathbf{u})}_n = \mathbf{H}\bar{\mathbf{m}}_n$, $\mathbf{P}^{(\mathbf{u})}_n = \mathbf{H}\bar{\mathbf{P}}_n\mathbf{H}^\top$ are given by filtering and smoothing. By combining the above properties we see that all the terms required for the natural gradient updates and hyperparameter learning can be computed efficiently in $\mathcal{O}(N_t M_\mathbf{s}^3 d_t^3)$. We call this approach the *spatio-temporal sparse variational GP* (ST-SVGP). The full algorithm is given in Alg. 1.

## 4   Further Improving the Temporal and Spatial Scaling

We now propose two approaches to further improve the computational properties of ST-VGP and ST-SVGP. First, we show how parallel filtering and smoothing can be used for non-conjugate GP inference, which results in a theoretical span complexity of $\mathcal{O}(\log N_t d^3)$. We then present a spatial mean-field approximation, which can be used independently, or in combination with sparsity.

### 4.1   Parallel Bayesian Filtering and Smoothing

The associative scan algorithm [9] uses a divide-and-conquer approach combined with parallelisation to convert $N$ sequential *associative* operations into $\log N$ sequential steps (for an operator $*$, associativity implies $(a*b)*c = a*(b*c)$). This algorithm has been made applicable to conjugate Markov

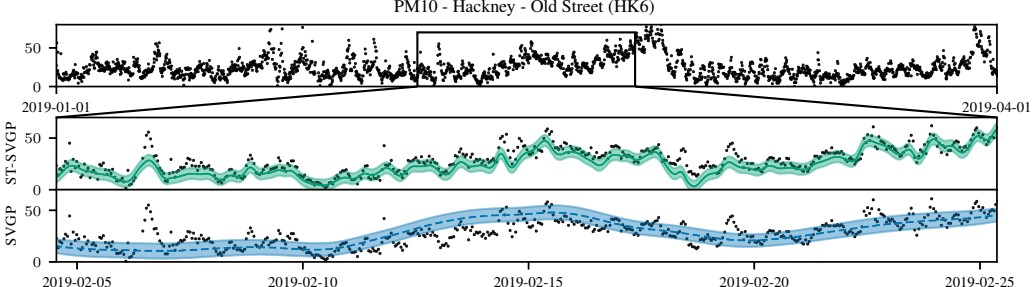

Figure 3: Observations of $PM_{10}$ at site HK6, showing rich short-scale structure (**top**). Mean and 95% confidence of ST-SVGP trained with 30 spatial inducing points (totalling 64,770 inducing points) and SVGP with 2000 inducing points (minibatch size 100). Both models have similar training times. ST-SVGP captures the complex structure of the time series whereas SVGP smooths the data.

GPs by deriving a new form of the Kalman filtering and smoothing operations that are associative [41, 13]. We give the form of these associative operators in App. E and show, for the first time, how these methods can be adapted to the non-conjugate setting. This follows directly from the use of the CVI approach to natural gradient VI, which requires only conjugate computations, *i.e.*, the linear filter and smoother. Consider the ST-SVGP ELBO:

$$\mathcal{L}_{\text{ST-SVGP}} = \underbrace{\mathbb{E}_{q(\mathbf{u})}\big[\mathbb{E}_{p(\mathbf{f}\,|\,\mathbf{u})}\big[\log p(\mathbf{Y}\,|\,\mathbf{f})\big]\big]}_{\substack{\text{factorises across time and space,}\\\text{compute in parallel}}} - \underbrace{\mathbb{E}_{q(\mathbf{u})}\big[\log \mathrm{N}(\widetilde{\mathbf{Y}}\,|\,\mathbf{u},\widetilde{\mathbf{V}})\big]}_{\substack{\text{factorises across time,}\\\text{compute in parallel}}} + \underbrace{\log \mathbb{E}_{p(\mathbf{u})}\big[\mathrm{N}(\widetilde{\mathbf{Y}}\,|\,\mathbf{u},\widetilde{\mathbf{V}})\big]}_{\text{compute with parallel filter}}.$$

The first two terms can be computed in parallel, since they decompose across time given the marginals $q(\mathbf{u}_n)$. The final term can be computed via the parallel filter, and the required marginals $q(\mathbf{u}_n)$ via the parallel smoother, which makes ST-SVGP a highly parallelisable algorithm. Alg. 4 gives the filtering and smoothing algorithm and Alg. 2 shows how this method can be used in place of the sequential filter when performing inference. One drawback of the parallel filter is that when the state dimension is large, many of the available computational resources may be consumed by the arithmetic operations involved in a single filtering step, and the logarithmic scaling may be lost. Fortunately, the spatial mean-field approximation presented in Sec. 4.2 helps to alleviate this issue. In App. E we provide more details on the method as well as a detailed examination of its practical properties.

## 4.2 Spatial Mean-Field Approximation

We reconsider the state space model for the spatio-temporal GP derived in Sec. 2.1, which has process noise $\mathbf{q}_n \sim \mathrm{N}(\mathbf{0}, \mathbf{K}_{\mathbf{SS}}^{(\mathbf{s})} \otimes \mathbf{Q}_n^{(t)})$. This Kronecker structure implies $N_{\mathbf{s}}$ independent processes are linearly mixed using spatial covariance $\mathbf{K}_{\mathbf{SS}}^{(\mathbf{s})}$. The linearity of this operation makes it possible to reformulate the model to include the mixing as part of the measurement, rather than the prior:

$$\bar{\mathbf{f}}(t_{n+1}) = \mathbf{A}_n\,\bar{\mathbf{f}}(t_n) + \mathbf{q}_n, \qquad \mathbf{Y}_n\,|\,\bar{\mathbf{f}}(t_n) \sim p(\mathbf{Y}_n\,|\,[\mathbf{C}_{\mathbf{SS}}^{(\mathbf{s})} \otimes \mathbf{H}^{(t)}]\,\bar{\mathbf{f}}(t_n)), \tag{17}$$

where $\mathbf{q}_n \sim \mathrm{N}(\mathbf{0}, \mathbf{I}_{N_{\mathbf{s}}} \otimes \mathbf{Q}_n^{(t)})$ and $\mathbf{C}_{\mathbf{SS}}^{(\mathbf{s})}$ is the Cholesky factor of $\mathbf{K}_{\mathbf{SS}}^{(\mathbf{s})}$ (see App. F for the derivation). This has the benefit that now both $\mathbf{A}_n$ and $\mathbf{Q}_n = \mathbf{I}_{N_{\mathbf{s}}} \otimes \mathbf{Q}_n^{(t)}$, are block diagonal, such that under the prior the latent processes are fully independent. This enables a mean-field assumption between the $N_{\mathbf{s}}$ latent posterior processes: $q(\bar{\mathbf{f}}(t)) \approx \prod_{k=1}^{N_{\mathbf{s}}} q(\bar{\mathbf{f}}_k(t))$, where $\bar{\mathbf{f}}_k(t)$ is the $d_t$-dimensional state corresponding to spatial point $\mathbf{S}_k$. This approximation enforces block-diagonal structure in the state covariance, such that matrix operations acting on the full state can be avoided. Dependence between the latent processes is still captured via the measurement model (likelihood), and our experiments show that this approach still approximates the true posterior well (see Sec. 5 and App. K.6) whilst providing significant computational gains when $N_{\mathbf{s}}$ is large.

## 5 Experiments

We examine the scalability and performance of ST-VGP and its variants. Throughout, we use a Matérn-$3/2$ kernel and optimise the hyperparameters by maximising the ELBO using Adam [32].

Table 1: NYC-CRIME (small) results. ST-SVGP = SVGP when $\mathbf{Z}$ is fixed.

| TRAIN $\mathbf{Z}$ | MODEL | RMSE | NLPD |
|---|---|---|---|
| $\times$ | ST-SVGP | $3.02 \pm 0.13$ | $1.72 \pm 0.04$ |
| $\times$ | SVGP | $3.02 \pm 0.13$ | $1.72 \pm 0.04$ |
| $\checkmark$ | ST-SVGP | $2.79 \pm 0.15$ | $1.64 \pm 0.04$ |
| $\checkmark$ | SVGP | $2.94 \pm 0.12$ | $1.65 \pm 0.05$ |

Table 2: Test performance on matching average run time in seconds for the NYC-CRIME (large) count dataset.

| MODEL | TIME (CPU) | TIME (GPU) | RMSE | NLPD |
|---|---|---|---|---|
| ST-SVGP | $20.86 \pm 0.46$ | $0.61 \pm 0.00$ | $2.77 \pm 0.06$ | $1.66 \pm 0.02$ |
| MF-ST-SVGP | $20.69 \pm 0.86$ | $0.32 \pm 0.00$ | $2.75 \pm 0.04$ | $1.63 \pm 0.02$ |
| SVGP-1500 | $12.67 \pm 0.11$ | $0.13 \pm 0.00$ | $3.20 \pm 0.14$ | $1.82 \pm 0.05$ |
| SVGP-3000 | $80.80 \pm 3.42$ | $0.45 \pm 0.01$ | $3.02 \pm 0.18$ | $1.76 \pm 0.05$ |

We use learning rates of $\rho = 0.01$, $\beta = 1$ in the conjugate case, and $\rho = 0.01$, $\beta = 0.1$ in the non-conjugate case. We use 5-fold cross-validation (*i.e.*, 80–20 train-test split), train for 500 iterations (except for AIR-QUALITY where we train for 300) and report RMSE, negative log predictive density (NLPD, see App. K.1) and average per-iteration training times on CPU and GPU. When using a GPU, the parallel filter and smoother are used.

**Synthetic Experiment** We construct 7 toy datasets with rich temporal structure but smooth spatial structure (see App. K.2) and varying size: $N_t = 10, 82, 155, 227, 300, 500, 1000$, and construct a $500 \times 100$ grid that serves as a test set for all cases. As the dataset size increases we expect the predictive performance of all methods to improve at the expense of run time. We compare against SKI and SVGP (see Sec. 1.1). Fig. 2a shows that whilst SVGP becomes infeasible for more than 300 time steps, the ST-SVGP variants scale linearly with time and are faster than SKI (except for the very small datasets, in which the model compile time in JAX dominates). In App. K we show the test performance of all models.

**NYC-CRIME – Count Dataset** We model crime numbers across New York City, USA (NYC), using daily complaint data from [1]. Crime data has seasonal trends and is spatially dependent. Accurate modelling can lead to more efficient allocation of police resources [20, 3]. We first consider a small subset of the data to highlight when our methods exactly recover SVGP. We bin the data from 1st to 10th of January 2014 ($N_t = 10$) into a spatial grid of size $30 \times 30$ and drop cells that do not intersect with land ($N_\mathbf{s} = 447$, $N = 4470$). We run ST-SVGP and SVGP with inducing points initialised to the same locations. We plot the training ELBO in Fig. 2b and performance in Table 1. For fixed inducing points, both models have the same training curve and provide the same predictions (up to numerical differences). Optimising the inducing points improves both methods. A comparable inference method for non-conjugate likelihoods has not yet been developed for SKI.

We next consider observations from January to July 2014, with daily binning ($N_t = 182$) and the same spatial grid ($N_\mathbf{s} = 447$, $N = 81,354$). We run ST-SVGP and its mean-field variant (MF-ST-SVGP) with 30 spatial inducing points (equivalent to SVGP with $M = 30 \times 182 = 5460$). Table 2 shows that our methods outperform SVGP (with $M = 1500$, $M = 3000$ and batch sizes 1500, 3000 respectively) because they can afford more inducing points for the same computational budget.

**Regression: AIR-QUALITY** Finally, we model $PM_{10}$ ($\mu g/m^3$) air quality across London, UK. The measurements exhibit periodic fluctuations and highly irregular behaviour due to events like weather and traffic jams. Using hourly data from the London air quality network [29] between January 2019 and April 2019 ($N_t = 2159$), we drop sensors that are not within the London boundaries or have more than 40% of missing data ($N_\mathbf{s} = 72$, $N = 155,448$). We run ST-SVGP and MF-ST-SVGP with 30 inducing points in space (equivalent to SVGP with $M = 30 \times 2159 = 64,770$ inducing points). To ensure the run times are comparable on both CPU and GPU, we run SVGP with 2000, 2500, 5000, and 8000 inducing points with mini-batch sizes of 600, 800, 2000, and 3000 respectively. We run SKI with $N_t$ temporal inducing points and 6 inducing points in each spatial dimension.

Table 3: AIR-QUALITY regression. ST-SVGP fits the fast-varying structure well, whereas SVGP smooths the data. Average run time and standard deviation in seconds shown for a single training step. ST-SVGP and MF-ST-SVGP use 30 spatial inducing points, equivalent to SVGP with $30 \times 2159 = 64,770$ inducing points. Number of inducing points chosen to make run time comparable.

| MODEL (BATCH SIZE) | TIME (CPU) | TIME (GPU) | RMSE | NLPD | |
|---|---|---|---|---|---|
| ST-SVGP | $16.79 \pm 0.63$ | $4.47 \pm 0.01$ | $\mathbf{9.96 \pm 0.56}$ | $\mathbf{8.29 \pm 0.80}$ | $\leftarrow$ full spatio-temporal model |
| MF-ST-SVGP | $13.74 \pm 0.49$ | $0.85 \pm 0.01$ | $10.42 \pm 0.63$ | $8.52 \pm 0.91$ | $\leftarrow$ with mean-field assumption |
| SVGP-2000 (600) | $20.21 \pm 0.28$ | $0.17 \pm 0.00$ | $15.46 \pm 0.44$ | $12.93 \pm 0.95$ | |
| SVGP-2500 (800) | $40.90 \pm 1.11$ | $0.25 \pm 0.00$ | $15.53 \pm 1.09$ | $13.48 \pm 1.85$ | |
| SVGP-5000 (2000) | — | $1.19 \pm 0.00$ | $14.20 \pm 0.44$ | $12.73 \pm 0.73$ | baselines |
| SVGP-8000 (3000) | — | $4.09 \pm 0.01$ | $13.83 \pm 0.47$ | $12.40 \pm 0.75$ | |
| SKI | $23.36 \pm 1.01$ | $3.61 \pm 0.01$ | $12.01 \pm 0.55$ | $10.32 \pm 0.79$ | |

Our methods significantly outperform the SVGP baselines because they can afford considerably more inducing points. As shown in Fig. 3 the SVGP drastically smooths the data, whereas ST-SVGP fits the short-term structure well. The mean-field approach is significantly more efficient, especially when using the parallel algorithm, but we do observe a slight reduction in prediction quality.

## 6    Conclusions and Discussion

We have shown that variational inference and spatio-temporal filtering can be combined in a principled manner, introducing an approach to GP inference for spatio-temporal data that maintains the benefits of variational GPs, whilst exhibiting favourable scaling properties. Our experiments confirm that ST-SVGP outperforms baseline methods because the effective number of inducing points grows with the temporal horizon, without introducing a significant computational burden. Crucially, this leads to improved predictive performance, because fast varying temporal information is captured by the model. We demonstrated how to apply parallel filtering and smoothing in the non-conjugate GP case, but our empirical analysis identified a maximum state dimension of around $d \approx 50$, after which the sub-linear temporal scaling is lost. However, our proposed spatial mean-field approach alleviates this issue somewhat, making the combined algorithm extremely efficient even when both the number of time steps and spatial points are large. The resemblance of our framework to the VGP approach suggests many potential extensions, such as multi-task models [4] and deep GPs [16]. We provide JAX code for all methods at `https://github.com/AaltoML/spatio-temporal-GPs`.

### 6.1    Limitations and Societal Impact

We believe our work takes an important step towards allowing sophisticated GP models to be run on both resource constrained CPU machines and powerful GPUs, greatly expanding the usability of such models whilst also reducing unnecessary consumption of resources. However, when using predictions from such methods, the limitations of the model assumptions and potential inaccuracies when using approximate inference should be kept in mind, especially in cases such as crime rate monitoring where actions based on biased or incorrect predictions can have harmful societal consequences.

## Acknowledgments and Disclosure of Funding

This work was supported by the Academy of Finland Flagship programme: Finnish Center for Artificial Intelligence (FCAI). Parts of this project were done while OH was visiting Finland as part of the FCAI–Turing initiative, supported by HIIT/FCAI. OH acknowledges funding from The Alan Turing Institute PhD fellowship programme. AS and WJW acknowledge funding from the Academy of Finland (grant numbers 324345 and 339730). NAL contributed under the NVIDIA AI Technology Center (NVAITC) Finland program. TD acknowledges support from EPSRC (EP/T004134/1), UKRI Turing AI Fellowship (EP/V02678X/1), and the Lloyd's Register Foundation programme on Data Centric Engineering. We acknowledge the computational resources provided by the Aalto Science-IT project and CSC – IT Center for Science, Finland.

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
