# Supplementary Material for
# Spatio-Temporal Variational Gaussian Processes

## A  Nomenclature

Table 4: Overview of notation. Vectors: bold lowercase. Matrices: bold uppercase.

| Symbol | Size | Description |
|---|---|---|
| $\mathbf{X}^{(st)}$ | $N_t \times N_\mathbf{s} \times D$ | Data input locations |
| $\mathbf{Y}^{(st)}$ | $N_t \times N_\mathbf{s}$ | Observations |
| $\mathbf{X}$ | $N \times D$ | Data input locations in matrix form. $N = N_t N_\mathbf{s}$ |
| $\mathbf{Y}$ | $N \times 1$ | Observations in vector form |
| $\boldsymbol{t}$ | $N_t \times 1$ | Vector of time steps |
| $\mathbf{S}$ | $N_\mathbf{s} \times D_\mathbf{s}$ | Spatial input locations. $D_\mathbf{s} = D - 1$ |
| $\mathbf{Z_t}$ | $M_t \times 1$ | Temporal inducing inputs (we set $\mathbf{Z_t} = \boldsymbol{t}$) |
| $\mathbf{Z_S}$ | $M_\mathbf{s} \times D_\mathbf{s}$ | Spatial inducing inputs |
| $\mathbf{K}_{tt}^{(t)}$ | $N_t \times N_t$ | Temporal kernel evaluated at temporal data points |
| $\mathbf{K}_{\mathbf{Z_t Z_t}}^{(t)}$ | $M_t \times M_t$ | Temporal kernel evaluated at temporal inducing points (we set $M_t = N_t$) |
| $\mathbf{K}_{\mathbf{SS}}^{(\mathbf{s})}$ | $N_\mathbf{s} \times N_\mathbf{s}$ | Spatial kernel evaluated at spatial data points |
| $\mathbf{K}_{\mathbf{Z_s Z_s}}^{(\mathbf{s})}$ | $M_\mathbf{s} \times M_\mathbf{s}$ | Spatial kernel evaluated at spatial inducing points |
| $\mathbf{K}_{\mathbf{Z_s Z_s}}^{-(\mathbf{s})}$ | $M_\mathbf{s} \times M_\mathbf{s}$ | $(\mathbf{K}_{\mathbf{Z_s Z_s}}^{(\mathbf{s})})^{-1}$ |
| $\mathbf{K}_{\mathbf{S}_k \mathbf{Z_s}}^{(\mathbf{s})}$ | $N_\mathbf{s} \times M_\mathbf{s}$ | Prior covariance between the spatial data points and spatial inducing points |
| $\mathbf{m}_n$ | $N_\mathbf{s} \times 1$ | Posterior mean at time $t_n$ |
| $\mathbf{P}_n$ | $N_\mathbf{s} \times N_\mathbf{s}$ | Posterior covariance at time $t_n$ |
| $\bar{\mathbf{f}}(t)$ | – | Gaussian distributed state, $\bar{\mathbf{f}}(t_n) \sim \mathrm{N}(\mathbf{s}(t_n) \,|\, \bar{\mathbf{m}}_n, \bar{\mathbf{P}}_n)$ |
| $\bar{\mathbf{m}}_n$ | $d \times 1$ | State mean at time $t_n$ |
| $\bar{\mathbf{P}}_n$ | $d \times d$ | State covariance at time $t_n$ |
| $\mathbf{A}_n$ | $d \times d$ | Discrete state transition model |
| $\mathbf{Q}_n$ | $d \times d$ | Discrete state process noise covariance |
| $\mathbf{H}$ | $1 \times d$ | State measurement model, $f(t) = \mathbf{H}\mathbf{s}(t)$ |
| $\mathbf{A}_n^{(t)}$ | $d_t \times d_t$ | Discrete state transition model for a single latent component (determined by $\kappa_t$) |
| $\mathbf{Q}_n^{(t)}$ | $d_t \times d_t$ | Discrete state process noise covariance for a single latent component (determined by $\kappa_t$) |
| $\mathbf{H}^{(t)}$ | $1 \times d_t$ | State measurement model for a single latent component (determined by $\kappa_t$) |
| $\boldsymbol{\lambda} = \{\boldsymbol{\lambda}^{(1)}, \boldsymbol{\lambda}^{(2)}\}$ | – | Natural parameters of the approximate posterior |
| $\widetilde{\boldsymbol{\lambda}} = \{\widetilde{\boldsymbol{\lambda}}^{(1)}, \widetilde{\boldsymbol{\lambda}}^{(2)}\}$ | – | Natural parameters of the approximate likelihood |
| $\boldsymbol{\eta}$ | – | Natural parameters of the prior |
| $\widetilde{\mathbf{Y}}$ | $N \times 1$ | Approximate likelihood mean |
| $\widetilde{\mathbf{V}}$ | $N \times N$ | Approximate likelihood covariance |

## B  Kronecker Identities

Assuming that the matrices conform and are invertible when required, the following properties hold:

$$(\mathbf{A} \otimes \mathbf{B})^{-1} = \mathbf{A}^{-1} \otimes \mathbf{B}^{-1} \tag{18}$$

$$\mathbf{A} \otimes (\mathbf{B} + \mathbf{C}) = \mathbf{A} \otimes \mathbf{B} + \mathbf{A} \otimes \mathbf{C} \tag{19}$$

$$(\mathbf{B} + \mathbf{C}) \otimes \mathbf{A} = \mathbf{B} \otimes \mathbf{A} + \mathbf{C} \otimes \mathbf{A} \tag{20}$$

$$(\mathbf{A} \otimes \mathbf{B})(\mathbf{C} \otimes \mathbf{D}) = (\mathbf{AC}) \otimes (\mathbf{BD}) \tag{21}$$

For further properties of kronecker products in the context of GP regression see Ch. 5 of Saatçi [39].

## C   Filtering and Smoothing Algorithms

**Algorithm 3** Sequential filtering and smoothing

**Input:** Likelihood: $\{\widetilde{\mathbf{Y}}, \widetilde{\mathbf{V}}\}$, Initial state: $\{\bar{\mathbf{m}}_0, \bar{\mathbf{P}}_0\}$, Model matrices: $\{\mathbf{A}, \mathbf{Q}, \mathbf{H}\}$

▷ *Run filter:*
**for** $n = 1 : N_t$ **do**
  ▷ *Filter predict:*
  $\tilde{\mathbf{m}}_n = \mathbf{A}_n \bar{\mathbf{m}}_{n-1}, \quad \bar{\mathbf{P}}_n = \mathbf{A}_n \bar{\mathbf{P}}_{n-1} \mathbf{A}_n^\top + \mathbf{Q}_n$
  $\mathbf{\Lambda}_n = \mathbf{H}\bar{\mathbf{P}}_n \mathbf{H}^\top + \widetilde{\mathbf{V}}_n, \quad \mathbf{W}_n = \bar{\mathbf{P}}_n \mathbf{H}^\top \mathbf{\Lambda}_n^{-1}$
  ▷ *Compute log likelihood:*
  $\ell_n = \log \mathrm{N}(\widetilde{\mathbf{Y}}_n \,|\, \mathbf{H}\bar{\mathbf{m}}_n, \mathbf{\Lambda}_n)$
  ▷ *Filter update:*
  $\bar{\mathbf{m}}_n = \tilde{\mathbf{m}}_n + \mathbf{W}_n(\widetilde{\mathbf{Y}} - \mathbf{H}\tilde{\mathbf{m}}_n)$
  $\bar{\mathbf{P}}_n = \bar{\mathbf{P}}_n - \mathbf{W}_n \mathbf{\Lambda}_n \mathbf{W}_n^\top$
**end for**

▷ *Run smoother:*
**for** $n = N_t - 1 : 1$ **do**
  $\mathbf{G}_n = \bar{\mathbf{P}}_n \mathbf{A}_{n+1} \bar{\mathbf{P}}_n^{-1}$
  $\mathbf{R}_{n+1} = \mathbf{A}_{n+1} \bar{\mathbf{P}}_n \mathbf{A}_{n+1}^\top + \mathbf{Q}_{n+1}$
  $\bar{\mathbf{m}}_n = \bar{\mathbf{m}}_n + \mathbf{G}_n(\bar{\mathbf{m}}_{n+1} - \mathbf{A}_{n+1}\bar{\mathbf{m}}_n)$
  $\bar{\mathbf{P}}_n = \bar{\mathbf{P}}_n + \mathbf{G}_n(\bar{\mathbf{P}}_{n+1} - \mathbf{R}_{n+1})\mathbf{G}_n^\top$
**end for**

▷ *Return marginals and log likelihood:*
**return** $q(\mathbf{f}_n) = \mathrm{N}(\mathbf{f}_n \,|\, \mathbf{H}\bar{\mathbf{m}}_n, \mathbf{H}\bar{\mathbf{P}}_n\mathbf{H}^\top), \; \forall n$
  $\ell = \sum_n \ell_n$

---

**Algorithm 4** Parallel filtering and smoothing

**Input:** Likelihood: $\{\widetilde{\mathbf{Y}}, \widetilde{\mathbf{V}}\}$, Initial state: $\{\bar{\mathbf{m}}_0, \bar{\mathbf{P}}_0\}$, Model matrices: $\{\mathbf{A}, \mathbf{Q}, \mathbf{H}\}$

▷ *Initialise filtering elements:*
$\mathbf{A}_0 = \mathbf{I}, \qquad \mathbf{Q}_0 = \bar{\mathbf{P}}_0$
**for** $n = 1 : N_t$ **in parallel do**
  $\mathbf{T}_n = \mathbf{H}\mathbf{Q}_{n-1}\mathbf{H}^\top + \widetilde{\mathbf{V}}_n,$
  $\mathbf{K}_n = \mathbf{Q}_{n-1}\mathbf{H}^\top \mathbf{T}_n^{-1},$
  $\mathbf{B}_n = \mathbf{A}_{n-1} - \mathbf{K}_n \mathbf{H}\mathbf{A}_{n-1},$
  $\hat{\mathbf{m}}_n = \mathbf{K}_n \widetilde{\mathbf{Y}}_n, \qquad \hat{\mathbf{P}}_n = \mathbf{Q}_{n-1} - \mathbf{K}_n \mathbf{H}\mathbf{Q}_{n-1},$
  $\boldsymbol{\phi}_n = \mathbf{A}_{n-1}^\top \mathbf{H}^\top \mathbf{T}_n^{-1} \widetilde{\mathbf{Y}}_n,$
  $\mathbf{J}_n = \mathbf{A}_{n-1}^\top \mathbf{H}^\top \mathbf{T}_n^{-1} \mathbf{H}\mathbf{A}_{n-1}$
**end for**
$\hat{\mathbf{m}}_1 = \bar{\mathbf{m}}_0 + \mathbf{K}_1(\widetilde{\mathbf{Y}}_1 - \mathbf{H}\bar{\mathbf{m}}_0)$

▷ *Run associative scan:*
$\hat{\mathbf{m}}, \hat{\mathbf{P}} = \textbf{associative\_scan}((\mathbf{B}, \hat{\mathbf{m}}, \hat{\mathbf{P}}, \boldsymbol{\phi}, \mathbf{J}), \overset{\text{filter}}{*})$
where operator $\overset{\text{filter}}{*}$ defined by Eq. (31) and Eq. (32)

▷ *Compute log likelihood:*
**for** $n = 1 : N_t$ **in parallel do**
  $\mathbf{\Lambda}_n = \mathbf{H}(\mathbf{A}_n \hat{\mathbf{P}}_{n-1} \mathbf{A}_n^\top + \mathbf{Q}_n)\mathbf{H}^\top$
  $\ell_n = \log \mathrm{N}(\widetilde{\mathbf{Y}}_n \,|\, \mathbf{H}\mathbf{A}_n \hat{\mathbf{m}}_{n-1}, \mathbf{\Lambda}_n)$
**end for**

▷ *Initialise smoothing elements:*
$\mathbf{E}_{N_t} = \mathbf{0}, \quad \bar{\mathbf{m}}_{N_t} = \hat{\mathbf{m}}_{N_t}, \quad \bar{\mathbf{P}}_{N_t} = \hat{\mathbf{P}}_{N_t}$
**for** $n = 1 : N_t - 1$ **in parallel do**
  $\mathbf{E}_n = \hat{\mathbf{P}}_n \mathbf{A}_n^\top (\mathbf{A}_n \hat{\mathbf{P}}_n \mathbf{A}_n + \mathbf{Q}_n)^{-1}$
  $\bar{\mathbf{m}}_n = \hat{\mathbf{m}}_n - \mathbf{E}_n \mathbf{A}_n \hat{\mathbf{m}}_n$
  $\bar{\mathbf{P}}_n = \hat{\mathbf{P}}_n - \mathbf{E}_n \mathbf{A}_n \hat{\mathbf{P}}_n$
**end for**

▷ *Run associative scan:*
$\bar{\mathbf{m}}, \bar{\mathbf{P}} = \textbf{associative\_scan}((\mathbf{E}, \bar{\mathbf{m}}, \bar{\mathbf{P}}), \overset{\text{smoother}}{*})$
where operator $\overset{\text{smoother}}{*}$ defined by Eq. (34) and Eq. (35)

▷ *Return marginals and log likelihood:*
**return** $q(\mathbf{f}_n) = \mathrm{N}(\mathbf{f}_n \,|\, \mathbf{H}\bar{\mathbf{m}}_n, \mathbf{H}\bar{\mathbf{P}}_n\mathbf{H}^\top), \; \forall n$
  $\ell = \sum_n \ell_n$

---

## D   Sparse Kronecker Decomposition

The input locations and inducing points are ordered such that they lie on a space time grid, and we define $\mathbf{X} = \mathrm{vec}(\mathbf{X})$. See Sec. 2 and App. B for details. We have assumed a separable kernel between space and time, hence we can decompose $\mathbf{K}$ as a Kronecker product:

$$\mathbf{K}_{\mathbf{XX}} = \mathbf{K}_{tt}^{(t)} \otimes \mathbf{K}_{\mathbf{SS}}^{(\mathbf{s})}, \tag{22}$$

$$\mathbf{K}_{\mathbf{XZ}} = \mathbf{K}_{\mathbf{X}_t \mathbf{Z}_t}^{(t)} \otimes \mathbf{K}_{\mathbf{SZ}_\mathbf{s}}^{(\mathbf{s})} = \mathbf{K}_{tt}^{(t)} \otimes \mathbf{K}_{\mathbf{SZ}_\mathbf{s}}^{(\mathbf{s})}, \tag{23}$$

$$\mathbf{K}_{\mathbf{ZZ}} = \mathbf{K}_{\mathbf{Z}_t \mathbf{Z}_t}^{(t)} \otimes \mathbf{K}_{\mathbf{Z}_\mathbf{s} \mathbf{Z}_\mathbf{s}}^{(\mathbf{s})} = \mathbf{K}_{tt}^{(t)} \otimes \mathbf{K}_{\mathbf{Z}_\mathbf{s} \mathbf{Z}_\mathbf{s}}^{(\mathbf{s})}. \tag{24}$$

The sparse conditional is,

$$p(\mathbf{f} \,|\, \mathbf{u}) = \mathrm{N}(\mathbf{f} \,|\, \mathbf{K}_{\mathbf{X},\mathbf{Z}} \mathbf{K}_{\mathbf{Z},\mathbf{Z}}^{-1} \mathbf{u}, \mathbf{K}_{\mathbf{X},\mathbf{X}} - \mathbf{K}_{\mathbf{X},\mathbf{Z}} \mathbf{K}_{\mathbf{Z},\mathbf{Z}}^{-1} \mathbf{K}_{\mathbf{Z},\mathbf{X}}), \tag{25}$$

which we can decompose using the Kronecker formulation of $\mathbf{K}$. Starting with the mean term:

$$
\begin{aligned}
\mu &= \mathbf{K_{XZ}K_{ZZ}^{-1}u} \\
&= \left[ (\mathbf{K}_{tt}^{(t)} \otimes \mathbf{K}_{\mathbf{SZ_s}}^{(\mathbf{s})})(\mathbf{K}_{tt}^{(t)} \otimes \mathbf{K}_{\mathbf{Z_sZ_s}}^{(\mathbf{s})})^{-1} \right] \mathbf{u} && \text{sub Eq. (23) and Eq. (24)} \\
&= \left[ (\mathbf{K}_{tt}^{(t)} \otimes \mathbf{K}_{\mathbf{SZ_s}}^{(\mathbf{s})})(\mathbf{K}_{tt}^{-(t)} \otimes \mathbf{K}_{\mathbf{Z_sZ_s}}^{-(\mathbf{s})}) \right] \mathbf{u} && \text{apply Eq. (18)} \\
&= \left[ (\mathbf{K}_{tt}^{(t)} \mathbf{K}_{tt}^{-(t)}) \otimes (\mathbf{K}_{\mathbf{SZ_s}}^{(\mathbf{s})} \mathbf{K}_{\mathbf{Z_sZ_s}}^{-(\mathbf{s})}) \right] \mathbf{u} && \text{apply Eq. (21)} \\
&= \left[ \mathbf{I} \otimes (\mathbf{K}_{\mathbf{SZ_s}}^{(\mathbf{s})} \mathbf{K}_{\mathbf{Z_sZ_s}}^{-(\mathbf{s})}) \right] \mathbf{u}, && (26)
\end{aligned}
$$

And now the covariance term:

$$
\begin{aligned}
\Sigma &= \mathbf{K_{XX}} - \mathbf{K_{XZ}K_{ZZ}^{-1}K_{ZX}} \\
&= (\mathbf{K}_{tt}^{(t)} \otimes \mathbf{K}_{\mathbf{SS}}^{(\mathbf{s})}) - (\mathbf{K}_{tt}^{(t)} \otimes \mathbf{K}_{\mathbf{SZ_s}}^{(\mathbf{s})})(\mathbf{K}_{tt}^{-(t)} \otimes \mathbf{K}_{\mathbf{Z_sZ_s}}^{-(\mathbf{s})})(\mathbf{K}_{tt}^{(t)} \otimes \mathbf{K}_{\mathbf{Z_sS}}^{(\mathbf{s})}) && \text{sub Eq. (22)–(24), Eq. (18)} \\
&= (\mathbf{K}_{tt}^{(t)} \otimes \mathbf{K}_{\mathbf{SS}}^{(\mathbf{s})}) - (\mathbf{K}_{tt}^{(t)} \mathbf{K}_{tt}^{-(t)} \mathbf{K}_{tt}^{(t)}) \otimes (\mathbf{K}_{\mathbf{SZ_s}}^{(\mathbf{s})} \mathbf{K}_{\mathbf{Z_sZ_s}}^{-(\mathbf{s})} \mathbf{K}_{\mathbf{Z_sS}}^{(\mathbf{s})}) && \text{apply Eq. (21)} \\
&= (\mathbf{K}_{tt}^{(t)} \otimes \mathbf{K}_{\mathbf{SS}}^{(\mathbf{s})}) - (\mathbf{K}_{tt}^{(t)}) \otimes (\mathbf{K}_{\mathbf{SZ_s}}^{(\mathbf{s})} \mathbf{K}_{\mathbf{Z_sZ_s}}^{-(\mathbf{s})} \mathbf{K}_{\mathbf{Z_sS}}^{(\mathbf{s})}) \\
&= \mathbf{K}_{tt}^{(t)} \otimes (\mathbf{K}_{\mathbf{SS}}^{(\mathbf{s})} - \mathbf{K}_{\mathbf{SZ_s}}^{(\mathbf{s})} \mathbf{K}_{\mathbf{Z_sZ_s}}^{-(\mathbf{s})} \mathbf{K}_{\mathbf{Z_sS}}^{(\mathbf{s})}). && \text{apply Eq. (19)} \quad (27)
\end{aligned}
$$

Substituting this back into the conditional we have:

$$
p(\mathbf{f} \,|\, \mathbf{u}) = \mathrm{N}(\mathbf{f} \,|\, \left[ \mathbf{I} \otimes (\mathbf{K}_{\mathbf{SS}}^{(\mathbf{s})} \otimes \mathbf{K}_{\mathbf{Z_sZ_s}}^{-(\mathbf{s})}) \right] \mathbf{u}, \mathbf{K}_{tt}^{(t)} \otimes (\mathbf{K}_{\mathbf{SS}}^{(\mathbf{s})} - \mathbf{K}_{\mathbf{SZ_s}}^{(\mathbf{s})} \mathbf{K}_{\mathbf{Z_sZ_s}}^{-(\mathbf{s})} \mathbf{K}_{\mathbf{Z_sS}}^{(\mathbf{s})})). \quad (28)
$$

At this point the covariance matrix is dense and so we cannot decompose any further.

# E  Parallel Filtering and Smoothing for Spatio-Temporal VGP

Here we provide more details of the parallel filtering and smoothing method for ST-SVGP as well as performance profiling and discussion of the benefits and drawbacks of the parallel and sequential approaches in practice.

Särkkä and García-Fernández [41] derive a new (equivalent) form of the linear filtering and smoothing operations that are *associative*. This property states that for an operator $*$ we have $(a*b)*c = a*(b*c)$. Associativity allows for application of the associative scan algorithm [9], which uses a divide-and-conquer approach to construct a computational 'tree', each level of which involves applying the operator $*$ to pairs of states in parallel before propagating the result up the tree. The height of this tree is $\log N_t$, and hence the computational span complexity is $\mathcal{O}(\log N_t)$ if there is enough parallel compute capacity to fully parallelise all of the required operations.

The associative filtering operator acts on a sequence of five elements $(\mathbf{B}_n, \hat{\mathbf{m}}_n, \hat{\mathbf{P}}_n, \boldsymbol{\phi}_n, \mathbf{J}_n)$, where the elements correspond to the parameters of the following quantities,

$$
\begin{aligned}
p(\bar{\mathbf{f}}_n \,|\, \bar{\mathbf{f}}_{n-1}, \mathbf{Y}_n) &= \mathrm{N}(\bar{\mathbf{f}}_n \,|\, \mathbf{B}_n \bar{\mathbf{f}}_{n-1} + \hat{\mathbf{m}}_n, \hat{\mathbf{P}}_n), \\
p(\mathbf{Y}_n \,|\, \bar{\mathbf{f}}_{n-1}) &= \mathrm{N}(\bar{\mathbf{f}}_{n-1} \,|\, \mathbf{J}_n \boldsymbol{\phi}_n, \mathbf{J}_n^{-1}).
\end{aligned}
\quad (29)
$$

$\boldsymbol{\phi}_n$, $\mathbf{J}_n$ are the precision-adjusted mean and precision of $p(\mathbf{Y}_n \,|\, \bar{\mathbf{f}}_{n-1})$. The elements are first initialised as follows (letting $\mathbf{A}_0 = \mathbf{I}$ and $\mathbf{Q}_0 = \mathbf{P}_\infty$),

$$
\begin{aligned}
\mathbf{T}_n &= \mathbf{H}\mathbf{Q}_{n-1}\mathbf{H}^\top + \mathbf{V}_n, \\
\mathbf{K}_n &= \mathbf{Q}_{n-1}\mathbf{H}^\top \mathbf{T}_n^{-1}, \\
\mathbf{B}_n &= \mathbf{A}_{n-1} - \mathbf{K}_n \mathbf{H} \mathbf{A}_{n-1}, \\
\hat{\mathbf{m}}_n &= \mathbf{K}_n \mathbf{Y}_n, \\
\hat{\mathbf{P}}_n &= \mathbf{Q}_{n-1} - \mathbf{K}_n \mathbf{H} \mathbf{Q}_{n-1}, \\
\boldsymbol{\phi}_n &= \mathbf{A}_{n-1}^\top \mathbf{H}^\top \mathbf{T}_n^{-1} \mathbf{Y}_n, \\
\mathbf{J}_n &= \mathbf{A}_{n-1}^\top \mathbf{H}^\top \mathbf{T}_n^{-1} \mathbf{H} \mathbf{A}_{n-1},
\end{aligned}
\quad (30)
$$

and we set $\hat{\mathbf{m}}_1 = \bar{\mathbf{m}}_0 + \mathbf{K}_1(\mathbf{Y}_1 - \mathbf{H}\bar{\mathbf{m}}_0)$ to account for the initial mean $\bar{\mathbf{m}}_0$. Here $\mathbf{A}_n$, $\mathbf{Q}_n$ and $\mathbf{H}$ are the model matrices defining the GP prior, as laid out in the main paper. Once initialised, the associative operator $\overset{\text{filter}}{*}$ is defined as,

$$(\mathbf{B}_{i,j}, \hat{\mathbf{m}}_{i,j}, \hat{\mathbf{P}}_{i,j}, \phi_{i,j}, \mathbf{J}_{i,j}) = (\mathbf{B}_i, \hat{\mathbf{m}}_i, \hat{\mathbf{P}}_i, \phi_i, \mathbf{J}_i) \overset{\text{filter}}{*} (\mathbf{B}_j, \hat{\mathbf{m}}_j, \hat{\mathbf{P}}_j, \phi_j, \mathbf{J}_j), \qquad (31)$$

where,

$$\begin{aligned}
\mathbf{W}_{i,j} &= (\hat{\mathbf{P}}_i^{-1} + \mathbf{J}_j)^{-1}\hat{\mathbf{P}}_i^{-1}, \\
\mathbf{B}_{i,j} &= \mathbf{B}_j\mathbf{W}_{i,j}\mathbf{B}_i, \\
\hat{\mathbf{m}}_{i,j} &= \mathbf{B}_j\mathbf{W}_{i,j}(\hat{\mathbf{m}}_i + \hat{\mathbf{P}}_i\phi_j) + \hat{\mathbf{m}}_j, \\
\hat{\mathbf{P}}_{i,j} &= \mathbf{B}_j\mathbf{W}_{i,j}\hat{\mathbf{P}}_i\mathbf{B}_j^\top + \hat{\mathbf{P}}_j, \\
\phi_{i,j} &= \mathbf{B}_i^\top\mathbf{W}_{i,j}^\top(\phi_j - \mathbf{J}_j\hat{\mathbf{m}}_i) + \phi_i, \\
\mathbf{J}_{i,j} &= \mathbf{B}_i^\top\mathbf{W}_{i,j}^\top\mathbf{J}_j\mathbf{B}_i + \mathbf{J}_i.
\end{aligned} \qquad (32)$$

The associative scan algorithm (which is implemented in various machine learning frameworks) is then applied to the operator defined by Eq. (31) and Eq. (32) to obtain the filtered elements, of which $\hat{\mathbf{m}}$ and $\hat{\mathbf{P}}$ correspond to the filtering means and covariances.

A similar approach leads to a parallel version of the Rauch-Tung-Striebel smoother by defining an associative operator which acts on the elements $(\mathbf{G}_n, \bar{\mathbf{m}}_n, \bar{\mathbf{P}}_n)$. The elements are initialised as,

$$\begin{aligned}
\mathbf{G}_n &= \hat{\mathbf{P}}_n\mathbf{A}_n^\top(\mathbf{A}_n\hat{\mathbf{P}}_n\mathbf{A}_n + \mathbf{Q}_n)^{-1}, \\
\bar{\mathbf{m}}_n &= \hat{\mathbf{m}}_n - \mathbf{G}_n\mathbf{A}_n\hat{\mathbf{m}}_n, \\
\bar{\mathbf{P}}_n &= \hat{\mathbf{P}}_n - \mathbf{G}_n\mathbf{A}_n\hat{\mathbf{P}}_n,
\end{aligned} \qquad (33)$$

for $n < N_t$ and we set $\mathbf{G}_{N_t} = \mathbf{0}$, $\bar{\mathbf{m}}_{N_t} = \hat{\mathbf{m}}_{N_t}$, $\bar{\mathbf{P}}_{N_t} = \hat{\mathbf{P}}_{N_t}$. Note that the initial value of $\mathbf{G}_n$ corresponds to the standard smoothing gain. The associative smoothing operator $\overset{\text{smoother}}{*}$ is defined as,

$$(\mathbf{G}_{i,j}, \bar{\mathbf{m}}_{i,j}, \bar{\mathbf{P}}_{i,j}) = (\mathbf{G}_i, \bar{\mathbf{m}}_i, \bar{\mathbf{P}}_i) \overset{\text{smoother}}{*} (\mathbf{G}_j, \bar{\mathbf{m}}_j, \bar{\mathbf{P}}_j), \qquad (34)$$

where

$$\begin{aligned}
\mathbf{G}_{i,j} &= \mathbf{G}_i\mathbf{G}_j, \\
\bar{\mathbf{m}}_{i,j} &= \mathbf{G}_i\bar{\mathbf{m}}_j + \bar{\mathbf{m}}_i, \\
\bar{\mathbf{P}}_{i,j} &= \mathbf{G}_i\bar{\mathbf{P}}_j\mathbf{G}_i^\top + \bar{\mathbf{P}}_i.
\end{aligned} \qquad (35)$$

After applying the associative scan to these elements using the operator defined by Eq. (34) and Eq. (35), $\bar{\mathbf{m}}$ and $\bar{\mathbf{P}}$ correspond to the smoothed means and covariances. The full filtering and smoothing algorithm is given in Alg. 4.

### E.1   Profiling the Parallel Filter and Smoother

Here we provide detailed analysis of ST-VGP applied to a spatial log-Gaussian Cox process where the bin widths can be altered to modify the number of temporal and spatial points.

Fig. 4 (a) plots the ratio of mean iteration wall-times obtained with the parallel and sequential filter approaches on a single NVIDIA Tesla V100 GPU. We can see that the parallel filter outperforms the sequential filter when the number of spatial points is $N_\mathbf{s} \leq 20$. In the low-dimensional case, where $N_\mathbf{s} = 10$ and the number of time steps $N_t = 3000$, the parallel filter achieves over 29x speed-up relative to the sequential filter. In contrast, the sequential filter outperforms the parallel filter when $N_\mathbf{s} \geq 50$. When $N_\mathbf{s} = 300$ and $N_t = 200$ the parallel filter is approximately $2\times$ slower than the sequential filter.

Fig. 4 (b) plots the ratio of mean iteration wall-times obtained with the sequential filter on a single Intel Xeon CPU E5-2698 v4 CPU and the parallel filter on a single NVIDIA Tesla V100 GPU. The parallel filter GPU runs outperform the CPU runs in all settings, with speed-ups as high as 16x for large $N_\mathbf{s}$. Surprisingly, higher speed-ups relative to the CPU runs are obtained with $N_\mathbf{s} = 10$ than with $N_\mathbf{s} = 20$. This suggest that the parallel filter is particularly efficient at very low $N_\mathbf{s}$.

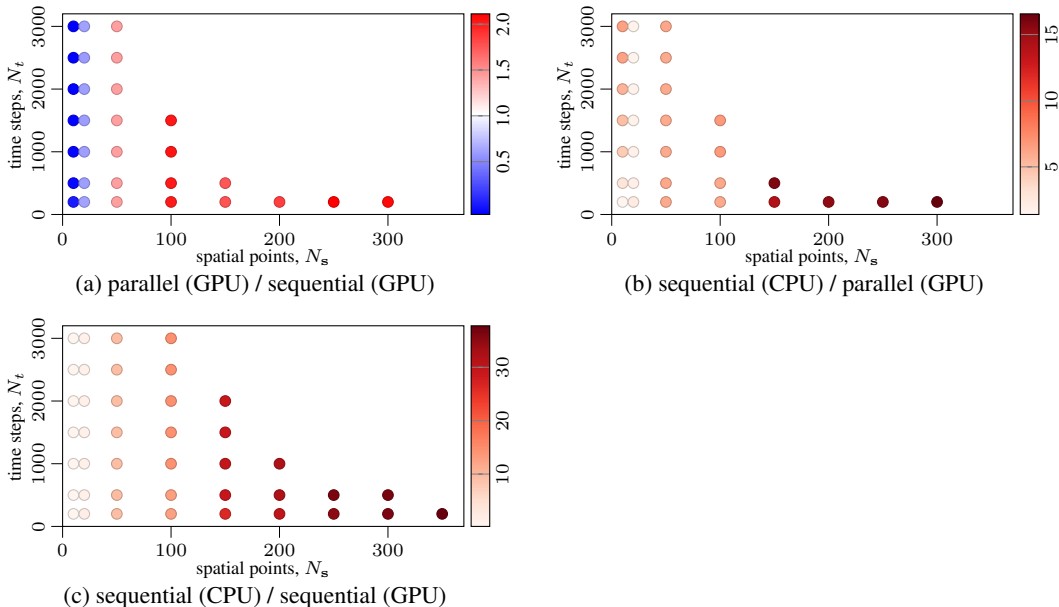

Figure 4: Comparison of relative runtime in seconds when running ST-VGP with the sequential filter/smoother on CPU and GPU and the parallel filter/smoother on GPU. The parallel algorithm outperforms the others when the number of spatial points is small, otherwise the sequential (GPU) method is best. The sequential (CPU) algorithm is competitive when 20 spatial points are used.

Fig. 4 (c) plots the ratio of mean iteration wall-times obtained with the sequential filter on a single Intel Xeon CPU E5-2698 v4 CPU and the sequential filter on a single NVIDIA Tesla V100 GPU. The sequential filter GPU runs outperform the CPU runs in all cases apart from those at $N_{\mathbf{s}} = 10$, with speed-up factors as has high as 37x. On the whole, these results suggest that the parallel filter is crucial to achieving good performance at low $N_{\mathbf{s}}$. However, it is not as effective as the sequential filter at high $N_{\mathbf{s}}$ and comes with increased memory footprint. Having both filtering options appears to be the best solution for a general-purpose spatio-temporal GP algorithm. All data associated with Fig. 4 are tabulated in Tables 5 to 7, where the dashes (—) denote configurations that run out of GPU memory.

In addition to the wall-time experiments, we also studied the characteristics of the parallel and sequential filtering approaches on GPUs via performance profiling. The primary reason causing the sequential filtering approach to be ill-suited for very low $N_{\mathbf{s}}$ when using a GPU is decreased computational intensity of operations. JAX is unable to fuse all operations in the algorithm and therefore, with low-dimensional data, the CUDA kernel execution times can become smaller than the kernel launch overheads. This issue is alleviated with increasing $N_{\mathbf{s}}$ as soon as the kernel executions take more time than kernel launches, since then the kernel launches can be completely overlapped with the execution of the previous kernel. With the parallel filtering approach, we are able to tackle the kernel launch overhead issue by introducing the associative scan operation, which JAX is able to combine as a single (but more computationally demanding) kernel. The reason the parallel filter is outperformed by the sequential filter at high $N_{\mathbf{s}}$ is more intricate. One of the reasons appears to be that the associative scan operation introduces more reads and writes to global GPU memory, and with high-dimensional data the operation becomes increasingly limited by memory bandwidth.

## F Reformulation of the Spatio-Temporal State Space Model for Spatial Mean-Field

The process noise covariance of the state $\bar{\mathbf{f}}(t)$ for a spatio-temporal GP is (see Sec. 2.1) $\mathbf{Q}_n = \mathbf{K}_{\mathbf{SS}}^{(\mathbf{s})} \otimes \mathbf{Q}_n^{(t)}$, where $\mathbf{K}_{\mathbf{SS}}^{(\mathbf{s})}$ is the spatial kernel evaluated at inputs $\mathbf{R}$, and $\mathbf{Q}_n^{(t)}$ is the process noise covariance of the state-space model induced by the temporal kernel. Similarly, the stationary distribution is given by $\mathbf{P}_\infty = \mathbf{K}_{\mathbf{SS}}^{(\mathbf{s})} \otimes \mathbf{P}_\infty^{(t)}$. Letting $\mathbf{C}_{\mathbf{SS}}^{(\mathbf{s})}$ be the Cholesky decomposition of $\mathbf{K}_{\mathbf{SS}}^{(\mathbf{s})}$,

Table 5: **Sequential (CPU)**: average training step time (secs) for sequential filter/smoother on CPU.

| | | SPATIAL POINTS, $N_{\mathbf{s}}$ | | | | | | | | |
|---|---|---|---|---|---|---|---|---|---|---|
| | | 10 | 20 | 50 | 100 | 150 | 200 | 250 | 300 | 350 |
| | 200 | 0.05 | 0.48 | 5.61 | 14.47 | 43.28 | 66.17 | 96.17 | 136.83 | 173.17 |
| | 500 | 0.12 | 0.97 | 14.23 | 36.93 | 121.43 | 176.67 | 252.40 | 344.02 | |
| | 1000 | 0.27 | 1.77 | 28.15 | 81.34 | 239.29 | 353.62 | | | |
| TIME STEPS, $N_t$ | 1500 | 0.40 | 2.54 | 41.69 | 122.86 | 357.36 | | | | |
| | 2000 | 0.52 | 3.32 | 56.24 | 164.59 | 475.51 | | | | |
| | 2500 | 0.65 | 4.06 | 70.62 | 203.48 | | | | | |
| | 3000 | 0.78 | 4.91 | 86.14 | 247.33 | | | | | |

Table 6: **Sequential (GPU)**: average training step time (secs) for sequential filter/smoother on GPU.

| | | SPATIAL POINTS, $N_{\mathbf{s}}$ | | | | | | | | |
|---|---|---|---|---|---|---|---|---|---|---|
| | | 10 | 20 | 50 | 100 | 150 | 200 | 250 | 300 | 350 |
| | 200 | 0.24 | 0.28 | 0.61 | 1.14 | 1.65 | 2.21 | 2.76 | 3.77 | 4.58 |
| | 500 | 0.62 | 0.70 | 1.52 | 2.87 | 4.16 | 5.53 | 6.92 | 9.43 | — |
| | 1000 | 1.19 | 1.40 | 3.05 | 5.71 | 8.15 | 11.05 | — | — | — |
| TIME STEPS, $N_t$ | 1500 | 1.86 | 2.11 | 4.58 | 8.60 | 12.37 | — | — | — | — |
| | 2000 | 2.44 | 2.79 | 6.11 | 11.47 | 16.38 | — | — | — | — |
| | 2500 | 3.01 | 3.55 | 7.62 | 14.26 | — | — | — | — | — |
| | 3000 | 3.56 | 4.13 | 9.16 | 17.03 | — | — | — | — | — |

Table 7: **Parallel (GPU)**: average training step time (secs) for parallel filter/smoother on GPU.

| | | SPATIAL POINTS, $N_{\mathbf{s}}$ | | | | | | | | |
|---|---|---|---|---|---|---|---|---|---|---|
| | | 10 | 20 | 50 | 100 | 150 | 200 | 250 | 300 | 350 |
| | 200 | 0.04 | 0.21 | 0.94 | 2.42 | 3.08 | 4.31 | 6.13 | 8.26 | – |
| | 500 | 0.04 | 0.50 | 2.35 | 6.07 | 7.68 | — | — | — | — |
| | 1000 | 0.06 | 1.00 | 4.70 | 12.19 | — | — | — | — | — |
| TIME STEPS, $N_t$ | 1500 | 0.08 | 1.49 | 7.05 | 18.28 | — | — | — | — | — |
| | 2000 | 0.09 | 2.00 | 9.39 | — | — | — | — | — | — |
| | 2500 | 0.10 | 2.49 | 11.77 | — | — | — | — | — | — |
| | 3000 | 0.12 | 2.99 | 14.13 | — | — | — | — | — | — |

we can use the Kronecker identities from App. B to rewrite the stationary state covariance as,

$$
\begin{aligned}
\mathbf{K}_{\mathbf{SS}}^{(\mathbf{s})} \otimes \mathbf{P}_{\infty}^{(t)} &= (\mathbf{K}_{\mathbf{SS}}^{(\mathbf{s})} \mathbf{I}_{N_{\mathbf{s}}}) \otimes (\mathbf{I}_{d_t} \mathbf{P}_{\infty}^{(t)}) \\
&= (\mathbf{K}_{\mathbf{SS}}^{(\mathbf{s})} \otimes \mathbf{I}_{d_t})(\mathbf{I}_{N_{\mathbf{s}}} \otimes \mathbf{P}_{\infty}^{(t)}) \\
&= (\mathbf{C}_{\mathbf{SS}}^{(\mathbf{s})} \otimes \mathbf{I}_{d_t})(\mathbf{I}_{N_{\mathbf{s}}} \otimes \mathbf{P}_{\infty}^{(t)})(\mathbf{C}_{\mathbf{SS}}^{(\mathbf{s})} \otimes \mathbf{I}_{d_t})^{\top},
\end{aligned}
\tag{36}
$$

and hence, recalling that the measurement model is $\mathbf{H} = \mathbf{I}_{N_{\mathbf{s}}} \otimes \mathbf{H}^{(t)}$, the prior covariance of the function, $\mathbf{f}_n = \mathbf{H}\mathbf{s}(t_n)$, is given by,

$$
\begin{aligned}
\mathrm{Cov}[\mathbf{f}_n] &= (\mathbf{I}_{N_{\mathbf{s}}} \otimes \mathbf{H}^{(t)})(\mathbf{K}_{\mathbf{SS}}^{(\mathbf{s})} \otimes \mathbf{P}_{\infty}^{(t)})(\mathbf{I}_{N_{\mathbf{s}}} \otimes \mathbf{H}^{(t)})^{\top} \\
&= (\mathbf{I}_{N_{\mathbf{s}}} \otimes \mathbf{H}^{(t)})(\mathbf{C}_{\mathbf{SS}}^{(\mathbf{s})} \otimes \mathbf{I}_{d_t})(\mathbf{I}_{N_{\mathbf{s}}} \otimes \mathbf{P}_{\infty}^{(t)})(\mathbf{C}_{\mathbf{SS}}^{(\mathbf{s})} \otimes \mathbf{I}_{d_t})^{\top}(\mathbf{I}_{N_{\mathbf{s}}} \otimes \mathbf{H}^{(t)})^{\top} \\
&= \underbrace{(\mathbf{C}_{\mathbf{SS}}^{(\mathbf{s})} \otimes \mathbf{H}^{(t)})}_{\text{Measurement model, } \mathbf{H}} \underbrace{(\mathbf{I}_{N_{\mathbf{s}}} \otimes \mathbf{P}_{\infty}^{(t)})}_{\mathbf{P}_{\infty}}(\mathbf{C}_{\mathbf{SS}}^{(\mathbf{s})} \otimes \mathbf{H}^{(t)})^{\top}.
\end{aligned}
\tag{37}
$$

We see from the above that the contribution from the spatial kernel can be included as part of the measurement model, $\mathbf{H}$, rather than the stationary state covariance. The process noise covariance $\mathbf{Q}_n$ can be deconstructed in a similar way (for stationary kernels, $\mathbf{Q}_n = \mathbf{P}_{\infty} - \mathbf{A}_n \mathbf{P}_{\infty} \mathbf{A}_n^{\top}$). Arguably, as discussed in Sec. 4.2, this presentation of the model is more intuitive since it becomes clear that $N_{\mathbf{s}}$ latent processes, each with an independent GP prior, are correlated via a measurement model in which the spatial covariance mixes the latent processes to generate the observations.

**Sparse Spatial Mean-Field** A similar argument holds for the sparse version of the model (ST-SVGP):

$$
\mathbf{K}_{\mathbf{Z_s Z_s}}^{(\mathbf{s})} \otimes \mathbf{P}_{\infty}^{(t)} = (\mathbf{C}_{\mathbf{Z_s Z_s}}^{(\mathbf{s})} \otimes \mathbf{I}_{d_t})(\mathbf{I}_{M_{\mathbf{s}}} \otimes \mathbf{P}_{\infty}^{(t)})(\mathbf{C}_{\mathbf{Z_s Z_s}}^{(\mathbf{s})} \otimes \mathbf{I}_{d_t})^{\top},
\tag{38}
$$

where $\mathbf{C}^{(\mathbf{s})}_{\mathbf{Z_s Z_s}}$ is the Cholesky factor of $\mathbf{K}^{(\mathbf{s})}_{\mathbf{Z_s Z_s}}$. Then,

$$
\begin{aligned}
\mathrm{Cov}[\mathbf{f}_n] &= ([\mathbf{K}^{(\mathbf{s})}_{\mathbf{SZ_s}} \mathbf{K}^{-(\mathbf{s})}_{\mathbf{Z_s Z_s}}] \otimes \mathbf{H}^{(t)})(\mathbf{K}^{(\mathbf{s})}_{\mathbf{Z_s Z_s}} \otimes \mathbf{P}^{(t)}_{\infty})([\mathbf{K}^{(\mathbf{s})}_{\mathbf{SZ_s}} \mathbf{K}^{-(\mathbf{s})}_{\mathbf{Z_s Z_s}}] \otimes \mathbf{H}^{(t)})^{\top} \\
&= ([\mathbf{K}^{(\mathbf{s})}_{\mathbf{SZ_s}} \mathbf{K}^{-(\mathbf{s})}_{\mathbf{Z_s Z_s}}] \otimes \mathbf{H}^{(t)})(\mathbf{C}^{(\mathbf{s})}_{\mathbf{Z_s Z_s}} \otimes \mathbf{I}_{d_t})(\mathbf{I}_{M_{\mathbf{s}}} \otimes \mathbf{P}^{(t)}_{\infty})(\mathbf{C}^{(\mathbf{s})}_{\mathbf{Z_s Z_s}} \otimes \mathbf{I}_{d_t})^{\top}([\mathbf{K}^{(\mathbf{s})}_{\mathbf{SZ_s}} \mathbf{K}^{-(\mathbf{s})}_{\mathbf{Z_s Z_s}}] \otimes \mathbf{H}^{(t)})^{\top} \\
&= \underbrace{([\mathbf{K}^{(\mathbf{s})}_{\mathbf{SZ_s}} \mathbf{C}^{-(\mathbf{s})}_{\mathbf{Z_s Z_s}}] \otimes \mathbf{H}^{(t)})}_{\text{Measurement model, } \mathbf{H}} \underbrace{(\mathbf{I}_{M_{\mathbf{s}}} \otimes \mathbf{P}^{(t)}_{\infty})}_{\mathbf{P}_{\infty}} ([\mathbf{K}^{(\mathbf{s})}_{\mathbf{SZ_s}} \mathbf{C}^{-(\mathbf{s})}_{\mathbf{Z_s Z_s}}] \otimes \mathbf{H}^{(t)})^{\top}. \quad (39)
\end{aligned}
$$

Again, this reformulation represents the same model as the standard form (and gives identical results), but enables the sparse and mean-field approximations to be combined since $\mathbf{P}_{\infty}$ is now block-diagonal.

## G  Exponential Family – Multivariate Gaussian Distribution

A Gaussian distribution $q(\mathbf{u}) = \mathrm{N}(\mathbf{u} \mid \mathbf{m}, \mathbf{P})$ is part of the exponential family with

$$\boldsymbol{\xi} = (\mathbf{m}, \mathbf{P}), \quad (40)$$

$$\boldsymbol{\lambda} = (\mathbf{P}^{-1}\mathbf{m}, -\frac{1}{2}\mathbf{P}^{-1}), \quad (41)$$

$$\boldsymbol{\mu} = (\mathbf{m}, \mathbf{m}\mathbf{m}^{\top} + \mathbf{P}), \quad (42)$$

where $\boldsymbol{\xi}$ are the moment parameters, $\boldsymbol{\lambda}$ are the natural parameters, and $\boldsymbol{\mu}$ are the expectation parameters. For further information see [47]. For completeness we provide a table to convert between the above parameterisations:

Table 8: Table of conversions between exponential family parameterisations

|  | $\to \boldsymbol{\xi}$ | $\to \boldsymbol{\lambda}$ | $\to \boldsymbol{\mu}$ |
|---|---|---|---|
| $\boldsymbol{\xi}$ | – | $[\boldsymbol{\xi}_2^{-1}\boldsymbol{\xi}_1, -\frac{1}{2}\boldsymbol{\xi}_2^{-1}]$ | $[\boldsymbol{\xi}_1, \boldsymbol{\xi}_1\boldsymbol{\xi}_1^T + \boldsymbol{\xi}_2]$ |
| $\boldsymbol{\lambda}$ | $[[-2\boldsymbol{\lambda}_2]^{-1}\boldsymbol{\lambda}_1, [-2\boldsymbol{\lambda}_2]^{-1}]$ | – | $[[-2\boldsymbol{\lambda}_2]^{-1}\boldsymbol{\lambda}_1, ([-2\boldsymbol{\lambda}_2]^{-1}\boldsymbol{\lambda}_1)^2 + [-2\boldsymbol{\lambda}_2]^{-1}]$ |
| $\boldsymbol{\mu}$ | $[\boldsymbol{\mu}_1, \boldsymbol{\mu}_2 - \boldsymbol{\mu}_1\boldsymbol{\mu}_1^T]$ | $[[\boldsymbol{\mu}_2 - \boldsymbol{\mu}_1\boldsymbol{\mu}_1^T]^{-1}\boldsymbol{\mu}_1, -\frac{1}{2}[\boldsymbol{\mu}_2 - \boldsymbol{\mu}_1\boldsymbol{\mu}_1^T]^{-1}]$ | – |

## H  Alternative Derivation of CVI Update Equations

We now show that after a natural gradient step the variational distribution can be decomposed as a conjugate Bayesian step with the model prior and an approximate likelihood.

### H.1  CVI Update

Applying the chain rule to the approximate likelihood parameters of the CVI update in Eq. (7) results in:

$$\widetilde{\boldsymbol{\lambda}}^{(1)}_{t+1} = (1 - \beta)\widetilde{\boldsymbol{\lambda}}^{(1)}_t + \beta \cdot (\mathbf{g}(\mathbf{m}) - 2\mathbf{g}(\mathbf{P})\mathbf{m}) \quad (43)$$

$$\widetilde{\boldsymbol{\lambda}}^{(2)}_{t+1} = (1 - \beta)\widetilde{\boldsymbol{\lambda}}^{(2)}_t + \beta \cdot \mathbf{g}(\mathbf{P}) \quad (44)$$

where, as defined in the main paper,

$$\mathbf{g}(\mathbf{m}) = \frac{\partial \mathbb{E}_q\left[\log p(\mathbf{Y} \mid \mathbf{f})\right]}{\partial \mathbf{m}} \quad (45)$$

$$\mathbf{g}(\mathbf{P}) = \frac{\partial \mathbb{E}_q\left[\log p(\mathbf{Y} \mid \mathbf{f})\right]}{\partial \mathbf{P}} \quad (46)$$

We now show that we recover Eq. (7) from standard natural gradients from [26]. Recall from Eq. (6) the natural gradient is given by

$$\boldsymbol{\lambda}_{t+1} = \boldsymbol{\lambda}_t + \beta \frac{\partial \mathcal{L}}{\partial \boldsymbol{\mu}} \quad (47)$$

and applying chain rule:

$$\boldsymbol{\lambda}_{t+1}^{(1)} = \boldsymbol{\lambda}_t^{(1)} + \beta\left(\frac{\partial \mathcal{L}}{\partial \mathbf{m}} - 2\frac{\partial \mathcal{L}}{\partial \mathbf{P}}\mathbf{m}\right) \tag{48}$$

$$\boldsymbol{\lambda}_{t+1}^{(2)} = \boldsymbol{\lambda}_t^{(2)} + \beta\left(\frac{\partial \mathcal{L}}{\partial \mathbf{P}}\right) \tag{49}$$

For ease of presentation we consider both natural parameters separately and for both will need the following result:

**Lemma H.1.** *Recursions of the form:*

$$R_{t+1} = (1-\beta)R_t + \beta b_t + \beta a \tag{50}$$

*where $R_1 = a$ can be rewritten as:*

$$R_{t+1} = \widetilde{R}_{t+1} + a \tag{51}$$

*where*

$$\widetilde{R}_{t+1} = (1-\beta)\widetilde{R}_t + \beta b_t \tag{52}$$

*with $\widetilde{R}_1 = 0$.*

*Proof.* The proof follows by induction. Using the fact that $R_1 = a$:

$$\begin{aligned}
R_2 &= (1-\beta)a + \beta b_1 + \beta a \\
&= (1-\beta)\widetilde{R}_1 + \beta b_1 + a \\
&= \widetilde{R}_2 + a
\end{aligned} \tag{53}$$

with $\widetilde{R}_2 = (1-\beta)\widetilde{R}_1 + \beta b_1$, and $\widetilde{R}_1 = 0$. In the next step:

$$\begin{aligned}
R_3 &= (1-\beta)R_2 + \beta b_2 + \beta a \\
&= (1-\beta)(\beta b_1 + a) + \beta b_2 + \beta a \\
&= (1-\beta)(\beta b_1) + \beta b_2 + a \\
&= (1-\beta)\widetilde{R}_2 + \beta b_2 + a \\
&= \widetilde{R}_3 + a
\end{aligned} \tag{54}$$

where $\widetilde{R}_3 = (1-\beta)\widetilde{R}_2 + \beta b_2$. In the general case:

$$\begin{aligned}
R_{t+1} &= (1-\beta)R_t + \beta b_t + \beta a \\
&= (1-\beta)(\widetilde{R}_t + a) + \beta b_t + \beta a \\
&= (1-\beta)\widetilde{R}_t + \beta b_t + a \\
&= \widetilde{R}_{t+1} + a
\end{aligned} \tag{55}$$

We have shown that the lemma holds on the first step and in a general step and so the proof holds by induction. $\square$

### H.1.1 First Natural Parameter

We first show that $\boldsymbol{\lambda}_{t+1}^{(1)}$ can be computed efficiently using the CVI updates. First; substituting Eqs. (45) and (46) into Eq. (48):

$$\begin{aligned}
\boldsymbol{\lambda}_{t+1}^{(1)} &= \boldsymbol{\lambda}_t^{(1)} + \beta\left(\mathbf{g}(\mathbf{m}) - \mathbf{K}^{-1}\mathbf{m} - 2\left[\mathbf{g}(\mathbf{P}) - \frac{1}{2}\left[-\mathbf{P}^{-1} + \mathbf{K}^{-1}\right]\right]\mathbf{m}\right) \\
&= \boldsymbol{\lambda}_t^{(1)} + \beta\left(\mathbf{g}(\mathbf{m}) - 2\mathbf{g}(\mathbf{P})\mathbf{m} - \mathbf{P}^{-1}\mathbf{m}\right)
\end{aligned} \tag{56}$$

Substituting $\boldsymbol{\lambda}_t^{(1)} = \mathbf{P}^{-1}\mathbf{m}$ and adding $\boldsymbol{\eta}^{(1)} = 0$:

$$\boldsymbol{\lambda}_{t+1}^{(1)} = (1-\beta)\boldsymbol{\lambda}_t^{(1)} + \beta\left(\mathbf{g}(\mathbf{m}) - 2\mathbf{g}(\mathbf{P})\mathbf{m}\right) + \beta\boldsymbol{\eta}^{(1)} \tag{57}$$

Applying Lemma H.1 we can directly rewrite the recursion as:

$$\boldsymbol{\lambda}_{t+1}^{(1)} = \widetilde{\boldsymbol{\lambda}}_t^{(1)} + \boldsymbol{\eta}^{(1)} \quad \text{where} \quad \widetilde{\boldsymbol{\lambda}}_t^{(1)} = (1-\beta)\widetilde{\boldsymbol{\lambda}}_{t-1}^{(1)} + \beta(\mathbf{g}(\mathbf{m}) - 2\mathbf{g}(\mathbf{P})\mathbf{m}) \tag{58}$$

and $\widetilde{\boldsymbol{\lambda}}_1^{(1)} = 0$ and $\boldsymbol{\lambda}_1^{(1)} = \boldsymbol{\eta}^{(1)} = 0$. This recovers the CVI update in Eq. (43).

### H.1.2 Second Natural Parameter

Following the steps for the first natural parameters we first substitute Eq. (46) into Eq. (49):

$$\boldsymbol{\lambda}_{t+1}^{(2)} = \boldsymbol{\lambda}_t^{(2)} + \beta \left( \mathbf{g}(\mathbf{P}) - \frac{1}{2} \left[ -\mathbf{P}^{-1} + \mathbf{K}^{-1} \right] \right)$$

$$= \boldsymbol{\lambda}_t^{(2)} + \beta \frac{1}{2} \mathbf{P}^{-1} + \beta \left( \mathbf{g}(\mathbf{P}) - \frac{1}{2} \mathbf{K}^{-1} \right) \tag{59}$$

substituting $\boldsymbol{\lambda}_t^{(2)} = -\frac{1}{2}\mathbf{P}^{-1}$ and $\boldsymbol{\eta}_t^{(2)} = -\frac{1}{2}\mathbf{K}^{-1}$:

$$\boldsymbol{\lambda}_{t+1}^{(2)} = (1-\beta)\boldsymbol{\lambda}_t^{(2)} + \beta \mathbf{g}(\mathbf{P}) + \beta \boldsymbol{\eta}_t^{(2)} \tag{60}$$

Applying Lemma H.1 the above simplifies to:

$$\boldsymbol{\lambda}_{t+1}^{(2)} = \widetilde{\boldsymbol{\lambda}}_t^{(2)} + \boldsymbol{\eta}^{(2)} \quad \text{where} \quad \widetilde{\boldsymbol{\lambda}}_t^{(2)} = (1-\beta)\widetilde{\boldsymbol{\lambda}}_{t-1}^{(2)} + \beta \mathbf{g}(\mathbf{P}) \tag{61}$$

and $\widetilde{\boldsymbol{\lambda}}_1^{(2)} = 0$ and $\boldsymbol{\lambda}_1^{(2)} = \boldsymbol{\eta}^{(2)}$. This recovers the CVI update in Eq. (44).

# I  Kronecker Structured Gaussian Marginals

The marginal $q(\mathbf{f}) = \int p(\mathbf{f} \,|\, \mathbf{u}) q(\mathbf{u}) \, \mathrm{d}\mathbf{u}$ is a Gaussian of the form:

$$q(\mathbf{f}) = \mathrm{N}(\mathbf{f} \,|\, \mathbf{A}\mathbf{m}^{(\mathbf{u})}, \mathbf{K}_{\mathbf{XX}} - \mathbf{A}\mathbf{K}_{\mathbf{ZX}} + \mathbf{A}\mathbf{P}^{(\mathbf{u})}\mathbf{A}^T) \quad \text{where} \quad \mathbf{A} = \mathbf{K}_{\mathbf{XZ}}\mathbf{K}_{\mathbf{ZZ}}^{-1}. \tag{62}$$

When $\mathbf{K}$ can be written as a Kronecker product the above can be simplified.

**Lemma I.1.** *Let* $\mathbf{K}_{\mathbf{XX}} = \mathbf{K}_{tt}^{(t)} \otimes \mathbf{K}_{\mathbf{SS}}^{(\mathbf{s})}$, $\mathbf{K}_{\mathbf{XZ}} = \mathbf{K}_{\mathbf{X}_t\mathbf{Z}_t}^{(t)} \otimes \mathbf{K}_{\mathbf{SZ_s}}^{(\mathbf{s})}$, $\mathbf{K}_{\mathbf{ZZ}} = \mathbf{K}_{tt}^{(t)} \otimes \mathbf{K}_{\mathbf{Z_sZ_s}}^{(\mathbf{s})}$ *then* $q(\mathbf{f})$
*can be decomposed as* $q(\mathbf{f}) = \mathrm{N}(\mathbf{f} \,|\, \mathbf{m}, \mathbf{P})$ *where,*

$$\mathbf{m} = \left[ \mathbf{I} \otimes (\mathbf{K}_{\mathbf{SZ_s}}^{(\mathbf{s})}\mathbf{K}_{\mathbf{Z_sZ_s}}^{-(\mathbf{s})}) \right] \mathbf{m}^{(\mathbf{u})} \tag{63}$$

$$\mathbf{P} = \left[ \mathbf{K}_{tt}^{(t)} \otimes \left( \mathbf{K}_{\mathbf{SS}}^{(\mathbf{s})} - \mathbf{K}_{\mathbf{SZ_s}}^{(\mathbf{s})}\mathbf{K}_{\mathbf{Z_sZ_s}}^{-(\mathbf{s})}\mathbf{K}_{\mathbf{Z_sS}}^{(\mathbf{s})} \right) \right] + \left[ \mathbf{I} \otimes \left( \mathbf{K}_{\mathbf{SZ_s}}^{(\mathbf{s})}\mathbf{K}_{\mathbf{Z_sZ_s}}^{-(\mathbf{s})} \right) \right] \mathbf{P}^{(\mathbf{u})} \left[ \mathbf{I} \otimes \left( \mathbf{K}_{\mathbf{Z_sZ_s}}^{-(\mathbf{s})}\mathbf{K}_{\mathbf{Z_sS}}^{(\mathbf{s})} \right) \right] \tag{64}$$

*Proof.* Starting with $\mathbf{m}$:

$$\mathbf{m} = \mathbf{K}_{\mathbf{XZ}}\mathbf{K}_{\mathbf{ZZ}}^{-1}\mathbf{m}^{(\mathbf{u})}$$

$$= \left[ (\mathbf{K}_{tt}^{(t)}\mathbf{K}_{tt}^{-(t)}) \otimes (\mathbf{K}_{\mathbf{SZ_s}}^{(\mathbf{s})}\mathbf{K}_{\mathbf{Z_sZ_s}}^{-(\mathbf{s})}) \right] \mathbf{m}^{(\mathbf{u})} \qquad \text{apply Eq. (21)}$$

$$= \left[ \mathbf{I} \otimes (\mathbf{K}_{\mathbf{SZ_s}}^{(\mathbf{s})}\mathbf{K}_{\mathbf{Z_sZ_s}}^{-(\mathbf{s})}) \right] \mathbf{m}^{(\mathbf{u})}. \tag{65}$$

And now dealing with $\mathbf{P}$. Let $G = \mathbf{K}_{\mathbf{XX}} - \mathbf{A}\mathbf{K}_{\mathbf{ZX}}$, $K = \mathbf{A}\mathbf{P}^{(\mathbf{u})}\mathbf{A}^T$. From the above we have shown that $\mathbf{K}_{\mathbf{XZ}}\mathbf{K}_{\mathbf{ZZ}}^{-1} = \left[ \mathbf{I} \otimes (\mathbf{K}_{\mathbf{SZ_s}}^{(\mathbf{s})}\mathbf{K}_{\mathbf{Z_sZ_s}}^{-(\mathbf{s})}) \right]$. First we substitute this into $G$:

$$G = \mathbf{K}_{\mathbf{XX}} - \left[ \mathbf{I} \otimes (\mathbf{K}_{\mathbf{SZ_s}}^{(\mathbf{s})}\mathbf{K}_{\mathbf{Z_sZ_s}}^{-(\mathbf{s})}) \right] \mathbf{K}_{\mathbf{ZX}}$$

$$= (\mathbf{K}_{tt}^{(t)} \otimes \mathbf{K}_{\mathbf{SS}}^{(\mathbf{s})}) - \left[ \mathbf{I} \otimes (\mathbf{K}_{\mathbf{SZ_s}}^{(\mathbf{s})}\mathbf{K}_{\mathbf{Z_sZ_s}}^{-(\mathbf{s})}) \right] \left[ \mathbf{K}_{tt}^{(t)} \otimes \mathbf{K}_{\mathbf{Z_sS}}^{(\mathbf{s})} \right]$$

$$= (\mathbf{K}_{tt}^{(t)} \otimes \mathbf{K}_{\mathbf{SS}}^{(\mathbf{s})}) - \left[ \mathbf{K}_{tt}^{(t)} \otimes (\mathbf{K}_{\mathbf{SZ_s}}^{(\mathbf{s})}\mathbf{K}_{\mathbf{Z_sZ_s}}^{-(\mathbf{s})}\mathbf{K}_{\mathbf{Z_sS}}^{(\mathbf{s})}) \right] \qquad \text{apply Eq. (21)}$$

$$= \mathbf{K}_{tt}^{(t)} \otimes \left( \mathbf{K}_{\mathbf{SS}}^{(\mathbf{s})} - \mathbf{K}_{\mathbf{SZ_s}}^{(\mathbf{s})}\mathbf{K}_{\mathbf{Z_sZ_s}}^{-(\mathbf{s})}\mathbf{K}_{\mathbf{Z_sS}}^{(\mathbf{s})} \right) \qquad \text{apply Eq. (19)} \tag{66}$$

And now substituting into $K$:

$$K = \mathbf{A}\mathbf{P}^{(\mathbf{u})}\mathbf{A}^T = \left[ \mathbf{I} \otimes (\mathbf{K}_{\mathbf{SZ_s}}^{(\mathbf{s})}\mathbf{K}_{\mathbf{Z_sZ_s}}^{-(\mathbf{s})}) \right] \mathbf{P}^{(\mathbf{u})} \left[ \mathbf{I} \otimes (\mathbf{K}_{\mathbf{SZ_s}}^{(\mathbf{s})}\mathbf{K}_{\mathbf{Z_sZ_s}}^{-(\mathbf{s})}) \right]^T$$

$$= \left[ \mathbf{I} \otimes (\mathbf{K}_{\mathbf{SZ_s}}^{(\mathbf{s})}\mathbf{K}_{\mathbf{Z_sZ_s}}^{-(\mathbf{s})}) \right] \mathbf{P}^{(\mathbf{u})} \left[ \mathbf{I} \otimes (\mathbf{K}_{\mathbf{Z_sZ_s}}^{-(\mathbf{s})}\mathbf{K}_{\mathbf{Z_sS}}^{(\mathbf{s})}) \right] \tag{67}$$

Combining $G, K$:

$$\mathbf{P} = G + K$$

$$= \mathbf{K}_{tt}^{(t)} \otimes \left( \mathbf{K}_{\mathbf{SS}}^{(\mathbf{s})} - \mathbf{K}_{\mathbf{SZ_s}}^{(\mathbf{s})}\mathbf{K}_{\mathbf{Z_sZ_s}}^{-(\mathbf{s})}\mathbf{K}_{\mathbf{Z_sS}}^{(\mathbf{s})} \right) + \left[ \mathbf{I} \otimes (\mathbf{K}_{\mathbf{SZ_s}}^{(\mathbf{s})}\mathbf{K}_{\mathbf{Z_sZ_s}}^{-(\mathbf{s})}) \right] \mathbf{P}^{(\mathbf{u})} \left[ \mathbf{I} \otimes (\mathbf{K}_{\mathbf{Z_sZ_s}}^{-(\mathbf{s})}\mathbf{K}_{\mathbf{Z_sS}}^{(\mathbf{s})}) \right] \tag{68}$$

which completes the proof. $\qquad\qquad\qquad\qquad\qquad\qquad\qquad\qquad\qquad\qquad\qquad\qquad\quad\square$

When the likelihood factorises across observations only the marginal $q(\mathbf{m}_{n,k})$ is required to compute the expected log likelihood. The marginal can be efficiently computed by utilising the fact that $\mathbf{I} \otimes (\mathbf{K}_{\mathbf{Z_s S}}^{(\mathbf{s})} \mathbf{K}_{\mathbf{Z_s Z_s}}^{-(\mathbf{s})})$ is block-diagonal, where there are $N_t$ blocks each of size $M_\mathbf{s} \times M_\mathbf{s}$.

**Lemma I.2.** *Following Lemma I.1 the marginal $q(\mathbf{f}_{n,k})$ is a Gaussian:* $q(\mathbf{f}_{n,k}) = \mathrm{N}(\mathbf{f}_{n,k} \mid \mathbf{m}_{n,k}, \mathbf{P}_{n,k})$ *where*

$$\mathbf{m}_{n,k} = \mathbf{K}_{\mathbf{S}_k \mathbf{Z_s}}^{(\mathbf{s})} \mathbf{K}_{\mathbf{Z_s Z_s}}^{-(\mathbf{s})} \mathbf{m}_n^{(\mathbf{u})} \tag{69}$$

$$\mathbf{P}_{n,k} = \sigma_t^2 \mathbf{K}_{\mathbf{X}_{n,k} \mathbf{X}_{n,k}}^{(\mathbf{s})} + \mathbf{K}_{\mathbf{S}_k \mathbf{Z_s}}^{(\mathbf{s})} \mathbf{K}_{\mathbf{Z_s Z_s}}^{-(\mathbf{s})} \left[ - \sigma_t^2 \mathbf{K}_{\mathbf{Z_s S}_k}^{(\mathbf{s})} + \mathbf{P}_n^{(\mathbf{u})} \mathbf{K}_{\mathbf{Z_s Z_s}}^{(\mathbf{s})} \mathbf{K}_{\mathbf{S}_k \mathbf{Z_s}}^{(\mathbf{s})} \right]. \tag{70}$$

*Proof.* To denote a single observation we subscript by $n,k$ and let $n$ denote the matrix/vector of elements at the $n$'th time step. First dealing with the mean:

$$\mathbf{m}_{n,k} = \left[ \left[ \mathbf{I} \otimes (\mathbf{K}_{\mathbf{SZ_s}}^{(\mathbf{s})} \mathbf{K}_{\mathbf{Z_s Z_s}}^{-(\mathbf{s})}) \right] \mathbf{m}^{(\mathbf{u})} \right]_{n,k}$$

$$= \left[ \mathbf{I} \otimes (\mathbf{K}_{\mathbf{SZ_s}}^{(\mathbf{s})} \mathbf{K}_{\mathbf{Z_s Z_s}}^{-(\mathbf{s})}) \right]_n \mathbf{m}_n^{(\mathbf{u})}$$

$$= \mathbf{K}_{\mathbf{S}_k \mathbf{Z_s}}^{(\mathbf{s})} \mathbf{K}_{\mathbf{Z_s Z_s}}^{-(\mathbf{s})} \mathbf{m}_n^{(\mathbf{u})}. \tag{71}$$

Where the second line holds due to $\mathbf{I} \otimes (\mathbf{K}_{\mathbf{Z_s S}}^{(\mathbf{s})} \mathbf{K}_{\mathbf{Z_s Z_s}}^{-(\mathbf{s})})$ being block diagonal and so each block affects separate elements of $\mathbf{m}^{(\mathbf{u})}$. The last line simply selects the relevant block from the block diagonal matrix. Deriving the form of the variance follows the same steps:

$$\mathbf{P}_{n,k} = [\mathbf{P}]_{n,k}$$

$$= \mathbf{K}_{\mathbf{X}_{n,k} \mathbf{X}_{n,k}}^{(t)} \cdot \left[ \mathbf{K}_{\mathbf{SS}}^{(\mathbf{s})} - \mathbf{K}_{\mathbf{SZ_s}}^{(\mathbf{s})} \mathbf{K}_{\mathbf{Z_s Z_s}}^{-(\mathbf{s})} \mathbf{K}_{\mathbf{Z_s S}}^{(\mathbf{s})} \right]_n + \left[ \mathbf{I} \otimes \left( \mathbf{K}_{\mathbf{SZ_s}}^{(\mathbf{s})} \mathbf{K}_{\mathbf{Z_s Z_s}}^{-(\mathbf{s})} \right) \right]_n \mathbf{P}_n^{(\mathbf{u})} \left[ \mathbf{I} \otimes \left( \mathbf{K}_{\mathbf{Z_s Z_s}}^{-(\mathbf{s})} \mathbf{K}_{\mathbf{Z_s S}}^{(\mathbf{s})} \right) \right]_n$$

$$= \sigma_t^2 \cdot \left[ \mathbf{K}_{\mathbf{X}_{n,k} \mathbf{X}_{n,k}}^{(\mathbf{s})} - \mathbf{K}_{\mathbf{S}_k \mathbf{Z_s}}^{(\mathbf{s})} \mathbf{K}_{\mathbf{Z_s Z_s}}^{-(\mathbf{s})} \mathbf{K}_{\mathbf{Z_s S}_k}^{(\mathbf{s})} \right] + \mathbf{K}_{\mathbf{S}_k \mathbf{Z_s}}^{(\mathbf{s})} \mathbf{K}_{\mathbf{Z_s Z_s}}^{(\mathbf{s})} \mathbf{P}_n^{(\mathbf{u})} \mathbf{K}_{\mathbf{Z_s Z_s}}^{(\mathbf{s})} \mathbf{K}_{\mathbf{S}_k \mathbf{Z_s}}^{(\mathbf{s})}$$

$$= \sigma_t^2 \mathbf{K}_{\mathbf{X}_{n,k} \mathbf{X}_{n,k}}^{(\mathbf{s})} + \mathbf{K}_{\mathbf{S}_k \mathbf{Z_s}}^{(\mathbf{s})} \mathbf{K}_{\mathbf{Z_s Z_s}}^{-(\mathbf{s})} \left[ - \sigma_t^2 \mathbf{K}_{\mathbf{Z_s S}_k}^{(\mathbf{s})} + \mathbf{P}_n^{(\mathbf{u})} \mathbf{K}_{\mathbf{Z_s Z_s}}^{(\mathbf{s})} \mathbf{K}_{\mathbf{S}_k \mathbf{Z_s}}^{(\mathbf{s})} \right] \tag{72}$$

which completes the proof. $\square$

## J  Block Diagonal Approximate Likelihood Natural Parameters

We now turn to the form of $\mathbf{g}(\mathbf{m})$ and $\mathbf{g}(\mathbf{P})$. The exact value of these can be easily calculated in any automatic differentiation library, but to use CVI we need to know where the non-zero elements of $\mathbf{g}(\mathbf{P})$ are.

**Lemma J.1.** *The form of $\mathbf{g}(\mathbf{P})$ is block diagonal where there are $N_t$ blocks each of size $M_\mathbf{s} \times M_\mathbf{s}$.*

*Proof.* The partial derivative of the expected log likelihood is:

$$\mathbf{g}(\mathbf{P}) = \frac{\partial \mathbb{E}_{q(\mathbf{f})} \left[ \log p(\mathbf{Y} \mid \mathbf{f}) \right]}{\partial \mathbf{P}^{(\mathbf{u})}} = \sum_n^{N_t} \sum_k^{N_\mathbf{s}} \frac{\partial \mathbb{E}_{q(\mathbf{f}_{n,k})} \left[ \log p(\mathbf{Y}_{n,k} \mid \mathbf{f}_{n,k}) \right]}{\partial \mathbf{P}^{(\mathbf{u})}}. \tag{73}$$

Applying chain rule:

$$\mathbf{g}(\mathbf{P}) = \sum_n^{N_t} \sum_k^{N_\mathbf{s}} \frac{\partial \mathbb{E}_{q(\mathbf{f}_{n,k})} \left[ \log p(\mathbf{Y}_{n,k} \mid \mathbf{f}_{n,k}) \right]}{\partial \mathbf{P}_{n,k}} \frac{\partial \mathbf{P}_{n,k}}{\partial \mathbf{P}^{(\mathbf{u})}}. \tag{74}$$

The first term is a scalar and so does not affect the final form. The second term is scalar-matrix derivative:

$$\frac{\partial \mathbf{P}_{n,k}}{\partial \mathbf{S}} = \begin{bmatrix} \frac{\partial \mathbf{P}_{n,k}}{\partial \mathbf{P}_{1,1}^{(\mathbf{u})}} & \frac{\partial \mathbf{P}_{n,k}}{\partial \mathbf{P}_{2,1}^{(\mathbf{u})}} & \cdots & \frac{\partial \mathbf{P}_{n,k}}{\partial \mathbf{P}_{M,1}^{(\mathbf{u})}} \\ \frac{\partial \mathbf{P}_{n,k}}{\partial \mathbf{P}_{1,2}^{(\mathbf{u})}} & \frac{\partial \mathbf{P}_{n,k}}{\partial \mathbf{P}_{2,2}^{(\mathbf{u})}} & \cdots & \frac{\partial \mathbf{P}_{n,k}}{\partial \mathbf{P}_{M,2}^{(\mathbf{u})}} \\ \vdots & \vdots & \ddots & \vdots \\ \frac{\partial \mathbf{P}_{n,k}}{\partial \mathbf{P}_{M,1}^{(\mathbf{u})}} & \frac{\partial \mathbf{P}_{n,k}}{\partial \mathbf{P}_{M,2}^{(\mathbf{u})}} & \cdots & \frac{\partial \mathbf{P}_{n,k}}{\partial \mathbf{P}_{M,M}^{(\mathbf{u})}} \end{bmatrix} \tag{75}$$

The inducing locations $\mathbf{Z} = ([\mathbf{Z}_n]_n^{N_t})$ are organised in time blocks and thus $\mathbf{P}^{(\mathbf{u})}$ is organised into time blocks. It is clear that only elements in Eq. (75) that correspond to the same time index as $n$ will be non-zero, because $\mathbf{P}_{n,k}$ only depends on $\mathbf{P}_n^{(\mathbf{u})}$. Due to the structure of $\mathbf{Z}$ these non-zero elements will be one of the $N_t \times N_t$ blocks on the block diagonal. The sum in Eq. (74) iterates over every $n, k$ and so the resulting matrix with have non-zero entries only in the block diagonal.

$\square$

## K  Further Details on Experiments

### K.1  Metrics Used

Let $\mathbf{Y} \in \mathbb{R}^{N \times 1}$ be the true value of the test data and $\boldsymbol{\mu} \in \mathbb{R}^{N \times 1}$, $\boldsymbol{\xi} \in \mathbb{R}^{N \times 1}$ be the predicted mean and variance, then we report,

$$\text{Root mean square error (RMSE)} = \sqrt{\frac{1}{N} \sum_{n=1}^{N} (\mathbf{Y}_n - \boldsymbol{\mu}_n)^2}, \tag{76}$$

$$\text{Negative log predictive density (NLPD)} = \frac{1}{N} \sum_{n=1}^{N} \log \int p(\mathbf{Y}_n \,|\, \mathbf{f}_n) \mathrm{N}(\mathbf{f}_n \,|\, \boldsymbol{\mu}_n, \boldsymbol{\xi}_n) \, \mathrm{d}\mathbf{f}_n. \tag{77}$$

With a Gaussian likelihood we make use of closed form solutions to the NLPD, otherwise we rewrite the NLPD as a LogSumExp function and approximate using quadrature with 100 points.

### K.2  Pseudo-periodic Functions

We construct toy datasets based on pseudo-periodic functions [30]:

$$\phi(t, c) = \sum_{i=3}^{7} \frac{1}{2^i} \sin \left( 2\pi \cdot (2^{2+i} + s_i) \cdot t \cdot c \right), \tag{78}$$

where $s_i$ are samples from a uniform distribution between $0$ and $2^i$ . These functions appear periodic but are never exactly repeating. The ground truth generative model is then defined as $f(t, r) = 50\phi(t, 3) \sin(4\pi r)$, with a Gaussian likelihood $y = f(t, r) + \varepsilon, \varepsilon \sim \mathrm{N}(0, 0.01)$.

### K.3  Computational Infrastructure

The experiments were run across various computational infrastructures.

**Run time experiment (Fig. 6)**  These experiments were run on an Intel Xeon E5-2698 v4 2.2 GHz CPU.

**Synthetic Experiment**  These experiments were run using 8, Intel Xeon CPU E5-2680 v4 @ 2.4 GHz, CPUs.

**Real World Experiments**  These experiments were run on an Intel Xeon Gold 6248 @ 2.5 GHz CPU or an NVIDIA Tesla V100 GPU.

### K.4  Baselines

We compare against two baselines, SVGP [26], SKI [52]:

**SVGP:**  We use the implementation provided in GPFlow [34].

**SKI:**  We use the implementation provided in GPyTorch [21]. We construct a *GridInterpolationKernel* and run with the default grid size or by matching the dimensions of the grid to the corresponding SVGP.

### K.5  Synthetic Experiment

For all models we use 6 spatial inducing points (or an equivalent grid of inducing points), with the spatial locations initialised through K-means. We initialise the likelihood noise to 0.1, use a Matérn-

$3/2$ kernel with lengthscale and variance of 0.1 and 1.0 respectively across all input dimensions, and run for 500 training iterations or one hour, whichever is shortest.

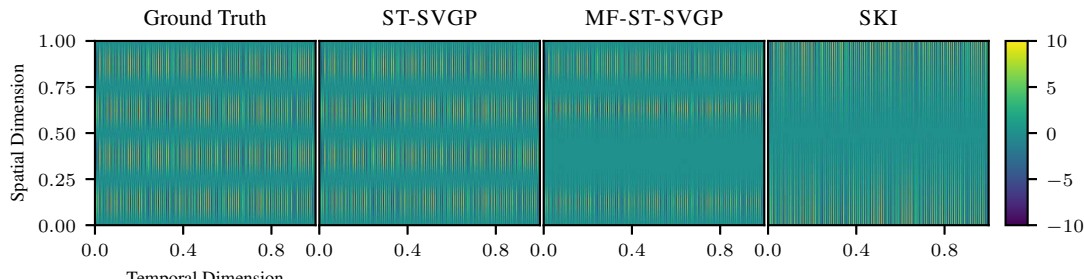

Figure 5: Test predictions on the synthetic experiment, dataset number 5. The ground truth (first panel) displays rich structure in the temporal dimension, but is smooth in the spatial dimension. Most of the models are able to capture some temporal structure but only ST-SVGP is able to accurately recover the ground truth.

We report the RMSE results on the synthetic experiment, detailed in the main paper, in Table 9. Each model and dataset combination is run five times with a different random seed for the data generation, and the reported means and standard deviations are calculated across these runs in Table 9. All experiments improve with the increasing training dataset size. The SVGP does not improve at the same rate as ST-SVGP because it very quickly reaches the one hour time limit and so is not trained beyond this. Fig. 5 shows the posterior predictive mean for various models on dataset number 5.

Table 9: Synthetic experiment: test RMSE. The size of the training data and its similarity to the test data increase with the dataset number. The mean and standard deviation across five runs is shown, with the data given by five random draws from the generative model given in Eq. (78).

| MODEL | 1 | 2 | 3 | 4 | 5 | 6 | 7 |
|---|---|---|---|---|---|---|---|
| ST-VGP | $4.86 \pm 0.38$ | $4.59 \pm 0.21$ | $4.42 \pm 0.29$ | $3.22 \pm 0.45$ | $2.49 \pm 0.07$ | $0.45 \pm 0.11$ | $0.85 \pm 0.03$ |
| SVGP | $4.95 \pm 0.38$ | $4.61 \pm 0.28$ | $4.30 \pm 0.45$ | $3.92 \pm 0.10$ | $3.78 \pm 0.25$ | $3.56 \pm 0.03$ | – |
| MF-ST-VGP | $4.91 \pm 0.38$ | $4.63 \pm 0.21$ | $4.52 \pm 0.24$ | $3.14 \pm 0.36$ | $2.71 \pm 0.19$ | $1.39 \pm 0.82$ | $2.13 \pm 0.04$ |
| SKI | $3.73 \pm 0.07$ | $3.69 \pm 0.03$ | $3.71 \pm 0.01$ | $3.57 \pm 0.07$ | $3.46 \pm 0.02$ | $3.34 \pm 0.01$ | $3.58 \pm 0.01$ |

### K.6   Comparison of Approximations, Fig. 6

In Fig. 6 we study the density of a single tree species, *Trichilia tuberculata*, from a $1000 \, \text{m} \times 500 \, \text{m}$ region of a rainforest in Panama [12]. We use a 5 m binning ($N_t = 200$) for the first spatial dimension (which we treat as time, $t$), and a varying bin size for the second spatial dimension (which we treat as space, $s$). The total number of data points is therefore $N = N_t N_s = 200 N_s$. We model the resulting count data use a log-Gaussian Cox process (approximated via a Poisson likelihood with an exponential link function). The spatio-temporal GP has a separable Matérn-$3/2$ kernel. The results show high-resolution binning is required to make accurate predictions on this dataset (the test NLPD falls as the number of spatial bins increases). Fig. 7 plots the data for this task, alongside the posterior mean given by the full model.

### K.7   Air Quality

For all models we initialise the likelihood noise to $5.0$, use a Matérn-$3/2$ kernel with lengthscales initialised to $[0.01, 0.2, 0, 2]$ and variance to $1.0$ and run for 300 epochs. See Fig. 3 for an example of the posterior obtained for a single spatial location over the course of three months.

### K.8   NYC-CRIME

For all models we use a Matérn-$3/2$ kernel with lengthscales initialised to $[0.001, 0.1, 0, 1]$ and variance to $1.0$ and run for 500 epochs. We use a natural gradient step size of $0.1$. See Fig. 1 for

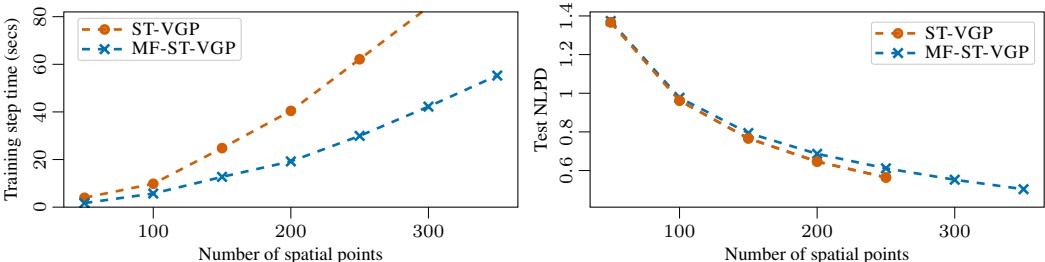

Figure 6: Comparison of ST-VGP and MF-ST-VGP. A two-dimensional grid of count data is binned with 200 time steps, $N_t$, and a varying number of spatial bins, $N_{\mathbf{s}}$. A Matérn-$3/2$ prior is used ($d_t = 2$, so $d = 2N_{\mathbf{s}}$). We show the time taken to perform one training step, averaged across 10 runs (**left**), and the test negative log predictive likelihood using 10-fold cross-validation (**right**).

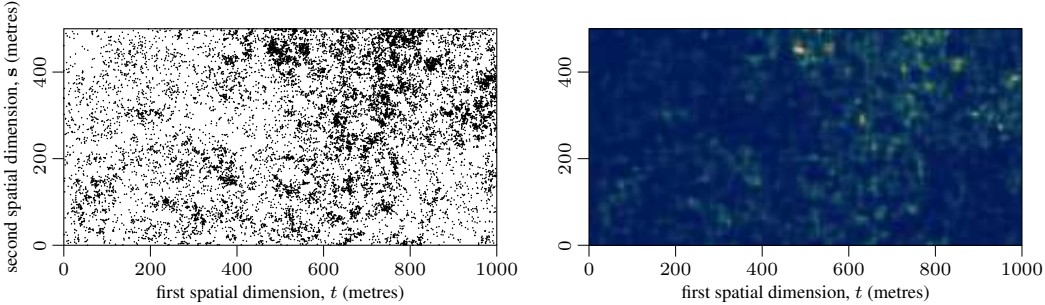

Figure 7: The tree count data used for the comparison of spatial approximations. The tree locations (**left**), are binned with temporal resolution $N_t = 200$ and spatial resolution $N_{\mathbf{s}} = 100$, and a log-Gaussian Cox process is applied. The posterior mean given by the full model (ST-VGP) is shown (**right**). See text for further details.

demonstrative plots of the predicted crime counts over NYC given by ST-VGP across eight days in 2015.

### K.9 Downloading Data

We have published the exact train-test folds for each dataset in Hamelijnck et al. [23].