# OpenReview forum: "Spatio-Temporal Variational Gaussian Processes"
_NeurIPS.cc/2021/Conference — NeurIPS 2021 Poster_

### Official Review · Reviewer_64f6 · 2021-06-28

**Rating:** 7
**Confidence:** 4

**Summary:**

The paper presents an approach for GP-Regression in the case of spatio-temporal data. The paper discusses a new scalable approach in the temporal domain by utilizing Bayesian filtering and smoothing in conjugation with conjugate-computation variational inference. Further scalability in the spatial domain is established by the use of an inducing point method with and without a mean field assumption.

**Main Review:**

All in all, I enjoyed reading the paper, and the approach seems like a nice addition to the computational library for GP-Regression.

Originality: The use of filtering and smoothing for GP-Regression is a well-known technique. Spatio-temporal GPs have been discussed in this context extensively as well. However, as the authors pointed out the scalable approach in the spatial domain is missing.

Quality: I think in general the paper is technically sound. However, when first reading the paper I was not so sure about it. The reason is that there are a lot of mathematical blunders and ambiguities, which should definitely be fixed. The claims are evaluated empirically, which is standard in this field of research. The method itself makes definitely sense in this context.

Clarity: I think this is one of the biggest weaknesses of the paper. Organization could definitely be improved and I oftentimes had a bit of a hard time following the discussed steps. But in general, I think the included background is informative and well selected. Though, I could see people having trouble understanding the state-space GP-regression when coming from the more machine-learning like perspective of function/ weight space GP-regression.

Significance: I think there is definitely a significance in this work, as GP-Regression is usually a bit problematic because of the scaling, though it is still used extensively in certain areas, such as Bayesian Optimization or for modelling dynamical systems in robotics.


•	Background: $f$ is a random function which describes a map e.g. $f:  \mathbb R \times \mathbb R^{D_s} \rightarrow \mathbb R$ and not a function of $\mathbf X \in \mathbb R^{N_t \times N_s \times D_s}$ as described in eq 1. At least, when one infers the inputs from the definitions of the kernel functions.

•	In general, the definitions are confusing and should be checked,e.g. check if $f_{n,k}=f(X_{n,k})$ is correct and properly define $X_{n,k}$.

•	The operator $\mathcal L_s$ is not mentioned in the text

•	2.1: The description of the process $\bar{f}$ is confusing as the relationship to the original process $f$ is established just at the end.

•	It would be helpful to add a bit more background on how the state space model is constructed from the kernel $\kappa_t(\cdot,\cdot)$, e.g. why it induces the dimensionality $d_t$ and also describe the limitations  that a finite dimensional SDE can only be established, when a suitable spectral decomposition of the kernel exist.

•	It should be mentioned that $p(y \mid H \bar{f}(t_n))$ has to be chosen Gaussian, as otherwise Kalman Filtering and Smoothing and CVI is not possible. Later on in the ELBOs this is assumed anyway.

•	$p(u)=N(u \mid 0, K_{zz})$ is a finite dimensional Gaussian not a Gaussian process and p(u) is not a random variable ($=$ not $\sim$).

•	The notations for the covariances, e.g. $K_{zz}$ are discussed in the appendix. I am fine with it; however, it should be referenced as I was confused in the beginning.

•	2.2: The $\log$ is missing for the Fisher information matrix.

•	The acronym CVI is used in the paragraph headline before the definition.

•	Some figures and tables are not referenced in the text, such as figure 1.

•	3.1: In line 173 the integration should be carried over $\bar{f}$ and not $s$, I guess?

•	I had a bit of a hard time establishing the connection $\mathbb E_{p(f)}[N(\tilde Y \mid f, \tilde{V})]=p(\tilde Y)$ which is the whole point why one part of the ELBO can be calculated using the Kalman filter. Adding this to a sentence to the text would have helped me a lot.

•	One question I had was that for computing the ELBO the matrix exponential is needed. When backpropagating the gradient for the hyper parameters, is this efficient? As I am used to using the adjoint method for computing gradients of the (linear) differential equation.

•	Reference to Appendix for the RMSE and NLPD metrics is missing.


**Time Spent Reviewing:**

8

---

> ### Author Response · Authors · 2021-08-09
> **Response to 64f6**
>
> We are very grateful for this careful analysis of the notation and mathematical presentation. We have made corrections based on every single one of the above technical points and we feel this has significantly improved the clarity and readability of the paper, particularly the background section. We recognise that there are a lot of technical concepts combined in the paper and have worked hard to fix inaccuracies and added additional pointers to the appendix where necessary to help the reader. We also provide the code for all methods, which we hope will help with understanding when needed.
>
> “more background on how the state space model is constructed from the kernel”: in the background section we provided some references to the relevant material, but we have now additionally added a reference to chapter 12 of [1]. Whilst we have omitted some details that would be nice to have, we feel that it is best not to over-crowd this section with too many of these details. However, if accepted, we will use the additional content page to add some more discussion around the form of the state and its relationship with the function f, which comes via the measurement matrix $H: f=H \bar{f}$. We will also add a comment stating that not all kernels can be formulated in this way exactly, but can be approximated accurately.
>
> On the computation of the matrix exponential for computing the ELBO: you are correct that this is required. For almost all of the most common kernels, this matrix exponential can actually be computed in closed form very cheaply. Implementations of this for many kernels are provided in the supplement. If the matrix exponential is not available in closed form then most ML frameworks such as JAX and tensorflow provide implementations based on an iterative numerical approach. In our experience, these are quite efficient, but they do scale cubically with the state dimension.
>
> [1] Särkkä and Solin (2020) “Applied Stochastic Differential Equations” Cambridge University Press.

---

> > ### Comment · Reviewer_64f6 · 2021-08-23
> > **My recommendation remains the same**
> >
> > Thank you for your answers. My concerns are addressed. My recommendation remains the same.

---

### Official Review · Reviewer_HABe · 2021-07-16

**Rating:** 7
**Confidence:** 3

**Summary:**

The authors propose a novel combination of existing strategies for scalable inference for spatio-temporal GPs. The techniques utilized were Kernel separability for convenient Kronecker structure, inference by filter-smoother in the  time domain, inducing point methods, and what appear to be 2 original contributions, a mean-field approximation to their spatial latent processes, and some computational strategies leveraging parallelization of the optimization. The authors make some camparisons to other methods in terms of computation time and accuracy.

**Limitations And Societal Impact:**

The authors have not addressed any limitation and potential negative societal impact.  They might consider how scalability reduces resource consumption.

**Main Review:**

Overall, the paper is well written and organized, although there are a number of minor details which I will list below that should to be addressed where the authors give insufficient details, introduce notation that is not explained (other than in the supplement), or possibly have a typo. Otherwise, the authors do a good job of a difficult task, synthesizing a wide range of GP literature to set the proper context for the their approach and to understand the basic advantages that the techniques confer.

My main complaint in terms of content is with respect to Section 5 (Experiments). The authors conduct a number of experiments, 1 with synthetic data and 2 on real data. An obvious contender with their methods is Wilson & Nickisch's SKIP method, which also makes use of a regular grid for the inducing points. The authors show that SKIP compares favorably in terms of wall-clock time but do not compare it's performance otherwise except for an example in the supplement, which is insufficient for this analysis. It would be better if the authors could include SKIP performance in the main-paper results.

Minor points
- Have the authors made a comparison with Hensmen et al.' variational Fourier feature method? This also seems to be an obvious contender with the author's approach.
- Line 66- What is D_s?
- Line 88 - "Where d_t is the dimension of the state-space model induced by \kappa_t..." Could the authors elaborate on this? Why should the dimension at each spatial point be larger than 1?
- Equation (4) The authors introduce notion in the superscripts of matrices that is not explained. Although they describe the notation in the supplement an explanation of the superscripts needs to appear in the main text as well.
- Lines 134-135 - "This shows that the natural gradient is completely characterized bu updates to an approximate likelihood" I'm sorry but you kind of lost be here. It's not obvious at all from this explanation why that the latter should follow from the former. Please explain.
- Line 136 - What do the superscripts on the \lambda 's mean?
- Equation (11) - What do the superscripts (t) on the A, K, and Q matrices mean? Please explain in the main text.
- line 249 - The authors are making a claim about a mean field approximation, which I would have expected to translate into factorization (i.e. product) of _densities_ over spatial points but the have written a product of functions over spatial points. Did I just miss something huge or is this a typo?


**Time Spent Reviewing:**

3

---

> ### Author Response · Authors · 2021-08-09
> **Response to HABe**
>
> Thank you for your detailed comments. We will address them in the order received.
>
> On SKIP: as you noticed, we only included SKIP in the synthetic regression experiments. In these experiments we found that SKIP has a less favourable wall clock time performance compared to ST-VGP. Additionally because SKIP uses an interpolation based approach to avoid matrix inversions it does not return the exact Gaussian posterior (unlike ST-VGP), which we found caused significant smoothing in the spatial dimension (see Fig 5 in the appendix). Furthermore a non-Gaussian, natgrad VI approach is not available making comparisons difficult. But we agree that on the Air Quality experiment SKI is a viable baseline and will include it for the camera ready submission.
>
> On VFF (please also see reply to Reviewer MX1g): Fourier features are dense in nature and typically most efficient in use cases where the input domain of interest is a compact subset of $\mathbb{R}^d$ (and $d$ rather small). VFF is typically not well suited for problems with a very long temporal dimension, especially if the characteristic length-scale of the prior over the temporal dimension is much shorter than the length of the data ($\ell \ll T$) as this would require impractically many inducing features. We will add discussion regarding this in the paper.
>
> $D_s$ is the dimensionality of the spatial points. For example, in the air pollution task we have 3 dimensions: the temporal dimension and 2 spatial dimensions (longitude & latitude), so $D_s=2$. We have clarified this in the paper.
>
> $d_t$ is the dimensionality of the state space model for a single spatial point. This is greater than 1 because the state typically contains the function value, $f$, as well as some of its derivatives. The function is obtained from the state via measurement matrix $H: f=H \bar{f}$. We did not want to over-crowd this section with too many details, however, we have now re-worded things, added more references, and moved the description of the relationship between $f$ and $\bar{f}$ to the start of this paragraph to make it much clearer.
>
> Regarding notation in the superscripts of matrices: we have now added a short explanation and made a much clearer link to the nomenclature table in the appendix.
>
> Regarding natgrad CVI being characterised by approx likelihood updates: apologies for the lack of clarity on this point. This follows from the fact that the prior parameters $\eta$ in Eq. (7) are fixed and known (for fixed hyperparameters). The only part of the gradient that changes across subsequent iterations is the contribution from the likelihood. Therefore the natgrad update essentially only amounts to updates to an approximate likelihood component. We have clarified this, modifying this sentence to reference Eq. (7).
>
> The superscripts on the lambdas refer to the first and second natural parameters: $\lambda^1$ and $\lambda^2$. This is explained in App. B, and we have added a short explanation and reference to the appendix in the main text.
>
> “superscripts (t) on the $A$, $K$, and $Q$ matrices”: this is to distinguish between, e.g., the $A$ that represents the transition matrix for the full spatio-temporal model, and $A^{(t)}$ which represents just a single component. This is an important distinction because we can write $A=I \otimes A^{(t)}$. We have clarified in the main text.
>
> Thank you for noticing our mistake regarding the mean-field factorisation. We have now rectified this.
>
> Regarding societal impact, we will add this discussion to the main paper: Many real-world processes evolve across space and time, for example the two case studies considered here: air pollution and city crime rates. We believe our work takes an important step toward allowing sophisticated GP models to be run on both resource constrained CPU machines as well as powerful GPUs, greatly expanding the useability of such models whilst also reducing unnecessary consumption of resources. We see opportunities to extend this work to expand the family of applicable kernels and to further improve computation performance. However, when using predictions from such methods, the limitations of the model prior assumptions, and potential inaccuracies when using approximate inference should be kept in mind, especially in cases such as crime rate monitoring, where actions based on biased or incorrect predictions can have harmful societal consequences.

---

### Official Review · Reviewer_MX1g · 2021-07-16

**Rating:** 7
**Confidence:** 3

**Summary:**

The paper presents a novel approach to variational inference in Gaussian Processes for spatio-temporal data (including non-gaussian likelihood), using state-space representation for linear scaling in the temporal dimension. The authors show how using natural gradients and conjugate-computation VI approach leads to decomposition of the ELBO which, together with linear-time filtering and smoothing of the state-space representation, results in significant computational gains.


The results are supported experimentally on one synthetic and 2 real datasets (conjugate and non-conjugate data). The proposed models outperform the baselines (sparse variational GP).

Overall this is a very solid paper.

**Limitations And Societal Impact:**

The limitations of the work are discussed sufficiently. Societal impact is not addressed due to the general theoretical nature of the work.

**Main Review:**


The problem of computational complexity in GP inference is still an important one and, as this work shows, there is still more room for improvement. This paper, while not groundbreaking, definitely provides substantial value to the community.

### Strengths:

The results of the paper are useful. The typical inducing point approach does not work well in the temporal dimension and the number of inducing points needed increases proportionally, still resulting in effectively cubic complexity. With this in mind, the linear complexity in the temporal dimension proposed in this work provides significant computational benefits.
The authors propose further improvements to the method via spatial mean-field approximation and a parallel version of the filtering algorithm.

The theoretical results are clear. I have not checked all the math thoroughly, but the results look correct. Detailed proof and derivations are given in the appendix.

The experimental results are thorough. Both RMSE and predictive likelihood are shown with error bars, the computation time is given and the experimental settings are described in detail.

The writing is very clear and easy to follow.


### Weaknesses:

Related work is focused on spatio-temporal modelling and missing some other relevant references. Notably, Vincent et al. [1] proposed doubly stochastic variational GP, merging the best of two main approaches to GP inference: variational methods and state-space representation, by introducing inducing states in the state-space representation. Since the submitted work addresses the same problem of making inference in GP more efficient, the connection to [1] should be explained.
Another possibly relevant line of work is the Variational Fourier Features approach [2]. Unlike typical inducing points, VFF would be suitable for capturing temporal dependencies. Some discussion on this is desirable.

Experimental comparison is only provided to the sparse GP. If the (missing) relevant work is directly applicable to spatio-temporal data, a comparison is needed.


The work is quite dense in theory, which resulted in placing important details in the supplement. Algorithms 3 and 4 are referred to in the text and in alg 2, but can only be found in the appendix. At least explicitly refer to the appendix when talking about alg 3, 4.



[1] Adam, Vincent, et al. "Doubly sparse variational Gaussian processes." International Conference on Artificial Intelligence and Statistics. PMLR, 2020.
APA

[2] Hensman, James, Nicolas Durrande, and Arno Solin. "Variational Fourier Features for Gaussian Processes." J. Mach. Learn. Res. 18.1 (2017): 5537-5588.

**Time Spent Reviewing:**

5

---

> ### Author Response · Authors · 2021-08-09
> **Response to MX1g**
>
> Thank you for your review, and in particular for your discussion of the missing references around doubly sparse VI and VFF.
>
> On doubly sparse VI: paper [1] is indeed relevant work, however in their case the “global” approach to natgrad VI is used, rather than CVI, and [1] only focuses on the sparse case. We see this work not as an alternative to our approach, but as a method that could be combined with our approach. This would lead to an algorithm that is potentially sparse in both time and space, separately. This is indeed interesting, and we will reference this in the Related Work section, and add discussion around future work in Sec. 6.
>
> On VFF: the same partially applies to [2], because Fourier features are known to be far from ideal in covering long (or possibly unbounded) spans over time. This is due to the nature of VFF to try to fill the domain of interest with inducing feature functions starting from the lowest frequency. This means that covering a long temporal stretch of data with high variability, would require impractically many inducing features. However, spatial VFFs could possibly be combined with our spatio-temporal viewpoint, which would open an interesting avenue for future work. We will use the additional space in the camera-ready submission to add discussion related to [1] and [2].
>
> We have modified the references to Algorithms 3 and 4.

---

> > ### Comment · Reviewer_MX1g · 2021-08-27
> > **Acknowledging author reply**
> >
> > Thank you for the reply.
> > Adding the references and some discussion of the related works [1] and [2] will definitely benefit the manuscript.

---

### Official Review · Reviewer_LdLE · 2021-07-17

**Rating:** 6
**Confidence:** 3

**Summary:**

This paper presents an efficient algorithm for variational inference of spatio-temporal processes using Gaussian processes as the variational family. The paper leverages so-called conjugate-computation variational inference to compute/represent distributions in the approximating variational family. By leveraging Markov, separability of the covariance kernel and prior sparsity assumptions on the spatio-temporal process, the paper demonstrates that the cubic computational complexity can be improved to linear in terms of the time variable.

**Ethical Concerns:**

None.

**Limitations And Societal Impact:**

The authors do not address the limitations of their method or any negative societal impact. This could be important since their example dataset is of crime statistics in NYC. How does the bias in this method affect any conclusions drawn from the method?

**Main Review:**

Overall I found the paper to be an acceptable contribution to the literature. The ideas are pretty standard, but used in a sensible way. In terms of originality, I would argue that some of the ideas, particularly around using smoothing to compute the posterior and natural gradients, are not new, though they have not been applied to the current setting. The paper is well written and reasonably clear. The computational speed up noted above should be of significance, particularly for large spatio-temporal data sets.

One thing I would ask the authors to improve upon is the exposition on the sparse priors, and in particular improve the description of the supporting/inducing points. I would also urge them to be more specific in how the SPDE is related to the GP prior (how is f related to \bar{f})? Also, provide a citation for why the SPDE reduces to an SDE in their setting.

**Time Spent Reviewing:**

4

---

> ### Author Response · Authors · 2021-08-09
> **Response to LdLE**
>
> Thank you for your considerate review. We agree that whilst the ideas of spatio-temporal filtering have been previously explored, our work fills an important gap in the literature, clearly demonstrating that in the separable kernel case we can significantly improve compute time without sacrificing accuracy or convergence rate relative to state of the art VI methods.
>
> On how the SPDE is related to the GP prior: we have restructured the background to clearly state that f is obtained by applying the measurement model H to the state: $f=H \bar{f}$. Typically $\bar{f}$ collects the derivatives of $f$, and $H=[1 ~ 0 ~ ... ~ 0]$.
>
> On why the SPDE reduces to an SDE: we have added a reference to [1] to this section. We attempted to explain this at the end of the paragraph with references to [2] and [3]. We have now re-worded this to make it very clear that this is possible due to the separability of the kernel between time and space, which induces conditional independence of the spatial points given the state.
>
> Regarding the sparse priors: are you referring to the background sections on FITC and SVGP, or the spatial sparsity in Sec. 3.3? For the background section, we have corrected some ambiguities and added a couple of references to the appendix, in particular to explain the matrix notation used in Sec. 2.1.
>
> Regarding societal impact, we will add this discussion to the main paper: Many real-world processes evolve across space and time, for example the two case studies considered here: air pollution and city crime rates. We believe our work takes an important step toward allowing sophisticated GP models to be run on both resource constrained CPU machines as well as powerful GPUs, greatly expanding the useability of such models whilst also reducing unnecessary consumption of resources. We see opportunities to extend this work to expand the family of applicable kernels and to further improve computation performance. However, when using predictions from such methods, the limitations of the model prior assumptions, and potential inaccuracies when using approximate inference should be kept in mind, especially in cases such as crime rate monitoring where actions based on biased or incorrect predictions can have harmful societal consequences.
>
> [1] Hartikainen, Riihimäki and Särkkä (2011). “Sparse Spatio-Temporal Gaussian Processes with General Likelihoods.” Proceedings of International Conference on Artificial Neural Networks.
>
> [2] Tebbut, Solin and Turner (2020) “Combining pseudo-point and state-space approximations for sum-separable Gaussian processes” In Advances in Approximate Bayesian Inference (AABI).
>
> [3] O’Hagan (1998) “A Markov property for covariance structures” Statistics research report.

---

### Decision · Program_Chairs · 2021-09-27

**Decision:**

Accept (Poster)

**Comment:**

The authors of this paper improve on the computational complexity of inferring spatiotemporal GPs. Specifically, via a combination of sparse priors, inducing points, and Markov separability they reduce a cubic cost to a linear cost. The reviewers all agree that the work is clear and stands to benefit the community. Moreover the few remarks raised by the reviewers seem to be readily addressed in revision. I therefore am happy to recommend this paper be accepted at NeurIPS.